

# Identification and ranking of volcanic tsunami hazard sources in Southeast Asia

Edgar U. Zorn[1], Aiym Orynbaikyzy[2], Simon Plank[2], Andrey Babeyko[1], Herlan Darmawan[3], Ismail Fata Robbany[2], Thomas R. Walter[1]

[1]German Research Centre for Geosciences GFZ, Telegrafenberg, 14473 Potsdam, Germany
[2]German Aerospace Center DLR, Münchenerstr. 20, 82234 Wessling, Germany
[3]Geophysics Study Program, Department of Physics, Faculty of Mathematics and Natural Sciences, Universitas Gadjah Mada, Sekip Utara, Bulaksumur, Yogyakarta, Indonesia

*Correspondence to*: Edgar U. Zorn (zorn@gfz-potsdam.de)

**Abstract.**

Tsunamis caused by large volcanic eruptions and flanks collapsing into the sea are major hazards for nearby coastal regions.
They often occur with little precursory activity, and are thus challenging to detect in a timely manner. This makes the pre-emptive identification of volcanoes prone to causing tsunamis particularly important, as it allows for better hazard assessment and denser monitoring in these areas. Here, we present a catalogue of potentially tsunamigenic volcanoes in Southeast Asia and rank these volcanoes by their tsunami hazard. The ranking is based on a Multicriteria Decision Analysis (MCDA) composed of five individually weighted factors impacting flank stability and tsunami hazard. The data is sourced
from geological databases, remote sensing data, historical volcano induced tsunami records and our topographic analyses, mainly considering the eruptive and tsunami history, elevation relative to the distance from the sea, flank steepness, hydrothermal alteration as well as vegetation coverage. Out of 131 analysed volcanoes, we found 19 with particularly high tsunamigenic hazard potential in Indonesia (Anak Krakatau, Batu Tara, Iliwerung, Gamalama, Sangeang Api, Karangetang, Sirung, Wetar, Nila, Ruang, Serua) and Papua New Guinea (Kadovar, Ritter Island, Rabaul, Manam, Langila, Ulawun,
Bam), but also in the Philippines (Didicas). While some of these volcanoes, such as Anak Krakatau, are well-known for their deadly tsunamis, many others on this list are lesser known and monitored. We further performed tsunami travel time modelling on these high-hazard volcanoes, which indicates that future events could affect large coastal areas in a short time. This highlights the importance of individual tsunami hazard assessment for these volcanoes, dedicated volcanological monitoring, and the need for increased preparedness on the potentially affected coasts.



## 1 Introduction

Tsunamis are among the deadliest hazards affecting coastal regions around the world. While most tsunamis are caused by tectonic earthquakes, tsunamis induced by volcanic sources account for ~6% of global tsunamis (NGDC, 2021). As a result, volcanic tsunamis are far less researched, but have resulted in many deadly events that were often unexpected and hit shores without warning. This is because volcanic tsunamis are low-probability, high-impact and hardly predictable black swan events, causing some 26% of all volcano induced fatalities since 1800 (Brown et al., 2017). Due to the very high density of active volcanoes near coastlines, Southeast (SE) Asia is one of the most prominent regions in the world for volcano induced tsunami events. The most well-known example is Krakatau volcano, Indonesia, where a tsunami caused by a major eruption in 1883 had an estimated death toll of ~36,000 people (Hamzah et al., 2000; Brown et al., 2017). In 2018, another tsunami by the same volcano killed 437 people due to the instability of its regrown volcano flank, which caused a sector collapse into the sea (Walter et al., 2019; Darmawan et al., 2020; Omira and Ramalho, 2020). In 1888, Ritter Island in Papua New Guinea experienced a similar catastrophic collapse, resulting in a tsunami with a death toll exceeding several hundred people (Ward and Day, 2003). Even without flank or edifice instability, volcanic eruptions can still cause deadly tsunamis by the expulsion of pyroclastic density currents (PDCs) into the sea. Such events repeatedly occurred at Awu volcano, Indonesia, in 1856, 1892 and 1913 with a cumulative ~4,500 fatalities (Hidayat et al., 2020). The recent explosive eruption of the Hunga Tonga-Hunga Ha'apai volcano near Tonga caused a tsunami affecting the entire Pacific Ocean, which travelled faster towards the coasts than was expected. While the mechanisms of this tsunami are not fully understood yet, reports suggest the interaction of acoustic and water gravity waves as a possible explanation (Somerville et al., 2022). This may be similar to airwaves attributed to tsunamis produced by Taal volcano, Philippines, in 1911 and 1965 (Paris et al., 2014 and references therein). These and many further examples emphasise that such potentially catastrophic tsunamis occur frequently and may pose a severe threat for coastal regions even hundreds of kilometres away from the source.

As historical databases reveal, the details of the tsunami triggering source are often a subject of debate. Even for the largest and deadliest volcanogenic tsunami, at Krakatau in 1883, discussed processes include caldera collapse, pyroclastic flows into the sea, sector collapse, explosion, or combinations thereof (e.g., Yokoyama, 1981; Francis, 1985; Nomanbhoy and Satake, 1995; Maeno et al., 2006). This highlights that there are multiple ways in which a volcano can cause a tsunami, which are considered a secondary volcanic hazard and result from eruptions or sector collapses of volcanic edifices (McGuire, 2006; Paris, 2015). Specifically, eruptions are known to generate tsunamis through large PDCs resulting from column or lava dome collapses entering the sea (Carey et al., 2000; Watts and Waythomas, 2003), through phreatic explosions when lava interacts with seawater (Smith and Shepherd, 1993; Belousov et al., 2000), and through the collapse of a lava delta when large amounts of lava construct unstable new land in the sea (Poland and Orr, 2014; Di Traglia et al., 2018). The formation of a caldera during particularly large eruptions is also known to generate tsunamis as large parts of the volcanic edifice are moving or collapsing (Maeno et al., 2006). On the other hand, a landslide or sector collapse (or lateral collapse) and resulting debris avalanche from a volcanic edifice may also occur in association with volcanic activity, i.e., flank instability triggered





through the intrusion of cryptodomes or dykes, earthquakes, explosions or loading of the flank with eruptive products (Lipman and Mullineaux, 1981; Siebert, 1984; Murray and Voight, 1996; McGuire, 2006; Romero et al., 2021). Flank

instability can also be gravitationally driven without current volcanic unrest or eruptive activity through a deep-seated and slow-moving décollement (van Wyk De Vries and Borgia, 1996) and may be seen as a precursory stage of a catastrophic flank collapse. Hydrothermal alteration may further facilitate gravitational instability by altering the chemical and structural composition of the volcanic flanks or basement (van Wyk De Vries and Borgia, 1996; Heap et al., 2013; Heap et al., 2021). However, the interrelation between gravitational and magmatic flank instability are still poorly understood and may strongly

depend on the local geologic setting and structural architecture of the volcano (Poland et al., 2017).

The inherent problem of volcanogenic tsunamis is the lack of warning time and quick response options because these tsunamis are not recognized by the early warning systems designed to detect tectonic earthquake events via seismic monitoring (Hanka et al., 2010; Lauterjung et al., 2010) and scenario-based modelling (Harig et al., 2020). This shortcoming also affected the 2018 tsunami originating from Anak Krakatau, where a warning system was in operation, but not designed

to correctly interpret the moderate-sized seismic energy as a tsunamigenic event (Annunziato et al., 2019). To compensate for this blind spot and improve local preparedness, a comprehensive understanding of the tsunami hazards posed by volcanic sources is required to identify the most likely individual volcanoes to produce such an event. Here, we present an in-depth analysis of the tsunami hazards by volcanoes in the Southeast Asian seas, including Indonesia, Papua New Guinea, the Philippines, and India. We create a comprehensive catalogue of potentially tsunamigenic volcanoes and further use this data

to create a point-based ranking and identify the most likely candidates for sourcing potentially catastrophic tsunamis in the future.

## 2 Methods

### 2.1 Morphological evaluation and catalogue

For creating the catalogue we considered all active volcanoes in the SE-Asian region (Fig. 1), here including India,

Indonesia, Papua New Guinea, and the Philippines, that were listed in the Global Volcanism Program (GVP) database (Global Volcanism Program, 2013) with a maximum distance of 20 km to the sea (with one exception: Peuet Sague, which has an uncertain historical tsunami associated it). Volcanoes further inland were not considered as mass movements from eruptions or flank/sector collapses are unlikely to exceed such a distance, although in some circumstances this may still be possible (Kieffer, 1981; Yoshida et al., 2012).




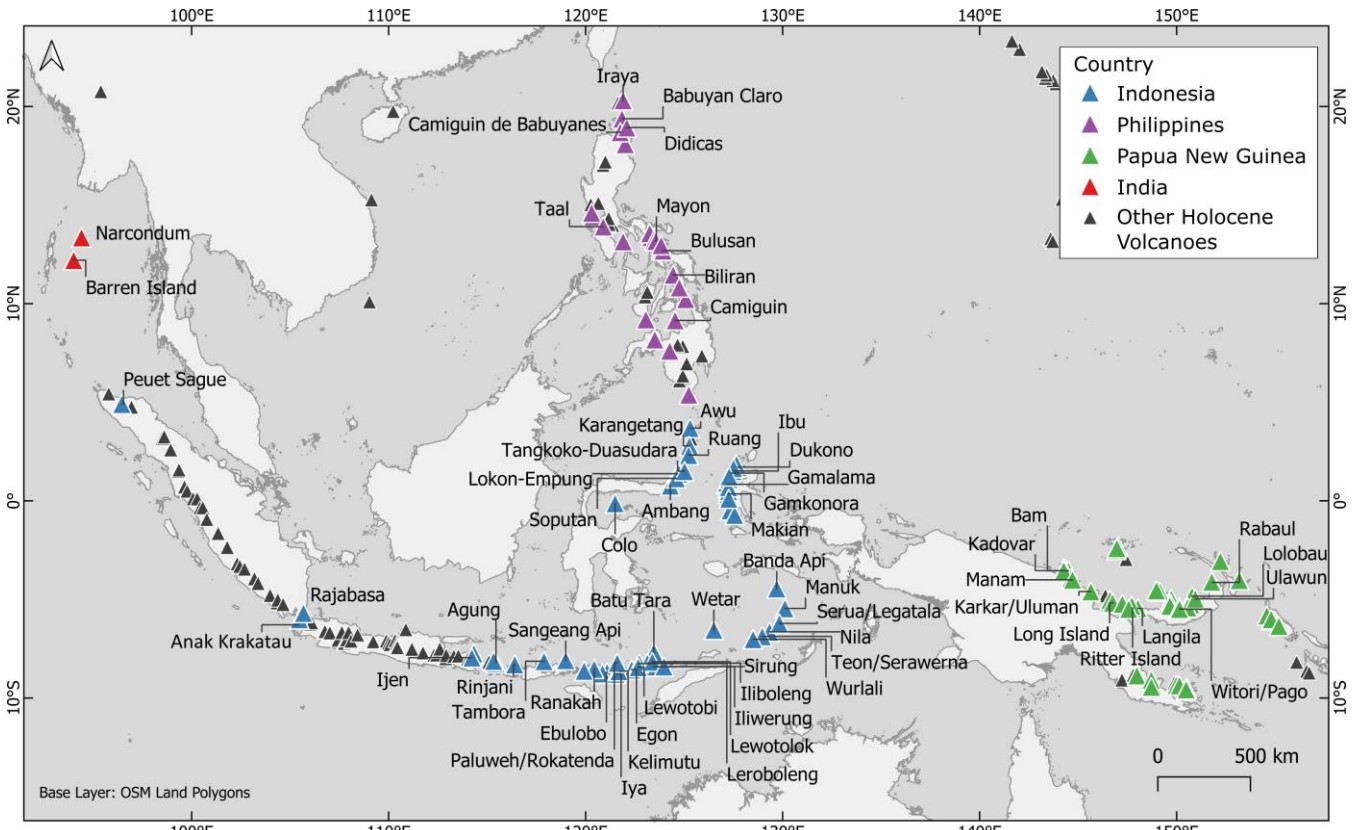

**Figure 1:** The overview map of all active volcanoes located in the SE-Asian region. Considered for our catalogue (in colour) are all subaerial volcanoes in Indonesia, India, Papua New Guinea and the Philippines at least 20 km away from the nearest coastline. The colour corresponds to the country where a volcano is located. In total, there are 214 active volcanoes between the four countries and 131 of them are considered in our catalogue. Base map data source: © OpenStreetMap contributors 2022. Distributed under the Open Data Commons Open Database License (ODbL) v1.0.

The catalogue contains a total of 131 volcanoes, colour-coded according to the country in Fig. 1. The information on the individual volcanoes was collected from databases, including the GVP (Global Volcanism Program, 2013) and the Global Historical Tsunami Database (NGDC, 2021), scientific literature, optical remote sensing data from Sentinel-2 satellites accessed via Sentinel Playground (ESA, 2022), as well as our own measurements using the Copernicus digital elevation model (DEM) (Airbus, 2020). Copernicus DEM is based on the previously available DEMs such as ASTER, SRTM90, SRTM30, SRTM30plus, GMTED2010, TerraSAR-X Radargrammetric DEM, and ALOS World 3D-30m.

We started by extracting morphometric data from the Copernicus DEMs using the NETVOLC (Euillades et al., 2013) and MORVOLC (Grosse et al., 2009; Grosse et al., 2012) codes and automatically delineated the edifice boundary of the



volcanoes. NETVOLC initiates with a starting location (usually the volcano summit) and a pixel range to draw the edifice boundary by iteratively looking for the minimum profile convexity. This is based on the assumption that volcanic edifices are bound by concave breaks in their slopes. Then, MORVOLC allowed us to use that boundary (and an optional crater outline, manually added) to collect data such as maximum elevation, slope steepness, or edifice volume for further

evaluation. For most volcanoes in our catalogue this approach worked very well. For some volcanoes (28/131 cases or ~21%), however, we had to delineate the edifice boundary manually as the automatic NETVOLC approach would produce no or visibly wrong boundaries. This is due to the more complex morphology of some volcanoes, which does not allow for a clear delineation of concave slope breaks.

The final catalogue (see supplementary material) includes (1) the coordinates of the volcano, (2) the volcano type, (3) its

activity status as the last eruption year, (4) how many tsunamis it has historically sourced, (5) its height and distance from the sea (as a ratio) as well as (6) the maximum average slope angle, (7) the possible collapse azimuth range (flanks facing the sea) and (8) the most likely azimuth for tsunamis (manually selected base on slope steepness or location relative to the sea), (9) the edifice volume, and (10) a summary of other hazardous features (see below).

**2.2 The volcanic tsunami-hazard ranking**

One key challenge in creating a *meaningful* ranking of the tsunami hazard posed by many individual volcanoes is the application of consistent criteria to allow for strong comparability. Since volcanoes have a large variety of shapes, morphologies and styles of activity, this is not a trivial task. We therefore only considered the data representing all volcanoes under the same conditions and we used as many objectively measurable criteria as possible to minimise human subjectivity

and inconsistency. Here, we decided on a Multicriteria Decision Analysis (MCDA), which is a method frequently used to aid in prioritising and decision-making for complex and multifaceted problems across many scientific disciplines. For example, in medical sciences evaluating the impact of drugs (Nutt et al., 2010), in Earth sciences assisting in the management of nuclear waste disposal sites (Morton et al., 2009), or in hazard evaluations of flood-prone sites (Fernández and Lutz, 2010; Rahmati et al., 2016; Toosi et al., 2019). Our MCDA system uses weighted point scores based on five major factors to

influence our final ranking. These factors were assigned (i) factor points based on their likelihood to contribute to the volcano's tsunami hazard, and (ii) factor weights based on how much the factor contributes to the volcano's tsunami hazard. By adding these weighed factor points to a final score we could rank the tsunami-hazard from the individual volcanoes:

$$Score = F_1 \cdot W_1 + F_1 \cdot W_1 + ... + F_n \cdot W_n \qquad (1)$$

where *F* represents the individual factor points from 0-100 (100 representing the highest tsunami-hazard) and *W* represents the

factor weight as a percentage (the total of all factor weights adding up to 100%). Due to the specific nature of some factors, the



full 100 points could not always be reached, but the point systems were designed to scale well between 0 and 100. This results in a final score that also has points between 0-100 and makes our ranking easier to comprehend and visualise.

We consider the following five factors and point systems for the ranking:

i)  **H/D-Ratio**: This is the height H of the volcanic edifice (i.e., the maximum elevation of the peak) as a fraction of the distance from that point to the sea D. To measure the height of volcano edifice, we used the 30 m Copernicus DEM GLO-30 (Airbus, 2020), which is considered the most reliable choice among the freely available and global DEMs (Guth and Geoffroy, 2021). Only for Anak Krakatau no post-collapse DEM was available, so we used a downsampled photogrammetric DEM previously published by Darmawan et al. (2020). We selected the maximum elevation of the edifice (extracted using the MORVOLC-code) and from that location measured the minimum distance to the shoreline using OpenStreetMap land polygons (OpenStreetMap, 2022). This parameter is highly relevant to the tsunami hazard of a volcano. Firstly, because a volcano with higher elevation can produce larger collapses and allow for more potential energy in mass movements, both from flank/sector/dome collapses and PDCs directed towards the sea. Secondly, if the mass movement source is closer to the sea it is more likely to actually reach it and produce a tsunami. The resulting values ranged from 0.02 to 0.89, so in order to convert the factor to the 0-100 point scale, the values were multiplied by 100. The only exception is Ritter Island, Papua New Guinea, which achieved a ratio of 1.77 due to the remnant collapse scar, which drops steeply towards the sea. We simplified this by assigning a maximum of 100 points.

ii) **Volcanic activity**: Frequent eruptive activity can exert strain on the flanks of a volcano as the volcano deforms, inflates and deflates as a result of pressurisation or the movement of intrusions. Additionally, erupted lava or tephra can quickly pile up and over steepen flanks, while constant high seismic activity can act as a trigger to mass movements. This means a volcano with a high activity level is far more likely to experience a sector/flank/dome collapse or produce pyroclastic flows compared to a quiescent one. While the exact mechanisms and timescales are generally not well understood, it is known that many volcanic systems have extended periods or cycles of quiescence or low eruptive frequency, followed by more frequent eruptions (e.g., Crisci et al., 1991; Gertisser and Keller, 2003; Turner, 2008). This suggests that a volcano that recently erupted is more likely to become more active or erupt again in the near-future compared to ones that have been quiet for a long time. Considering all above, we based our factor points on the time since the last known eruption of the volcano, with the data sourced from the GVP (Global Volcanism Program, 2013). However, due to the different timescales of eruptive cycles and the decreasing certainty of historic records, we decided to apply a non-linear point scale. Any volcano that erupted since the beginning of 2020 received a full 100 points. Every year before 2020 resulted in one point deduction and from the year 2000 only every decade is worth one point less, but to a minimum of 20 points as long as the volcano erupted within the Holocene. Similarly, if these



volcanoes experienced historical unrest episodes but no eruption, they were assigned 20 points. For volcanoes
where Holocene eruptions are presumed or considered likely, we assigned 10 points and 0 points were assigned
for volcanoes that are presumed to be extinct.

iii) **Tsunamigenic history**: Some volcanoes have an increased tendency to produce tsunamis, either through
frequent large eruptions or inherently unstable flanks. Here, we counted the number of tsunamis known from
historic records sourced from the Global Historical Tsunami Database (NGDC, 2021) and relevant review
papers (Hamzah et al., 2000; Paris et al., 2014; Mutaqin et al., 2019; Hidayat et al., 2020). Our compilation is
shown in table 1. We further considered signs of previous edifice instability that include collapse scars
(amphitheatres) and known submarine debris avalanche deposits. We assigned 10 points for every known
historical tsunami, 10 for a collapse scar or submarine deposit and 20 if the volcano has multiple scars or
submarine deposits.

iv) **Slope angle**: The steeper an edifice is the less stable it becomes. Typically, the angle of repose for natural
volcanic rocks lies between 30-40° and edifices exceeding this angle may experience gravitational instability.
Here, we measured the maximum average slope angle that is part of the MORVOLC output, meaning the
steepest part of the edifice calculated as the average slope value of the 50 m elevation interval of the edifice
with the highest mean average slope, see Grosse et al. (2009). However, we limited the output to only those
flanks facing towards the sea and excluded slopes facing inland. For the factor points, we doubled the steepness
value in degrees, so a 50° slope would equal the maximum of 100 points. We also note that this factor is a
major distinction from the H/D-ratio since it considers real measured local steepness. This may differ strongly
from the H/D-Ratio, e.g., if the volcano is very steep, but the summit is located far from the sea.

v) **Hazardous Features**: This factor is deliberately designed to encompass a broad collection of features
impacting the likelihood of a tsunami-generating event at the volcanoes. These features cannot be easily
quantified and have to be arbitrarily evaluated by a human and are easily missed or misinterpreted. Thus, in
order to minimise user bias and avoid unrealistic points, all listed features below were combined into one
factor. The data was collected by visually examining the Copernicus DEM GLO-30 (Airbus, 2020) as well as
optical satellite imagery from Sentinel-2 satellites accessed via Sentinel Playground (ESA, 2022). The
considered features include:

- Underwater extent: Whether a volcano and its flanks are submerged in the sea plays a very important role
when assessing tsunami hazards as an edifice failure or an erupting vent may be located partially or fully
underwater. We assigned 10 points if the edifice is partially submerged (so at least part of the edifice
outline used in the NETVOLC and MORVOLC codes reaches into the sea) and 20 points if the edifice is
fully in the water and all flanks reach into the sea

- Morphological features: Breached craters highlight a volcano's tendency to experience partial instability
and provide an easy path for eruption products to reach the sea. We assigned 15 points to volcanoes with




breached craters. Similarly, calderas signal that one or many large eruptions have occurred at this volcano and were also given 15 points. Finally, a larger volcanic edifice may have multiple secondary edifices and vents, adding further potential source locations to volcanogenic tsunamis. We thus added 15 points if one or more secondary peaks/vents were found.


- Vegetation: Dense vegetation and plant roots can significantly enhance a flank's stability and make a flank/sector collapse less likely (Gonzalez-Ollauri and Mickovski, 2017). Contrary, a flank's stability may significantly decrease due to the vegetation loss after major eruptions (Korup et al., 2019). Here, we visually inspected the volcano flanks using Sentinel-2 data (10 m spatial resolution). We gave no points if the volcano flanks were densely vegetated, 5 points if the volcano had vegetation free portions (most commonly, this is near the crater as a result of recent eruptions), and 15 points if at least one flank between summit and sea was free of vegetation.


- Hydrothermal alteration: Fumarole systems and weathering may significantly alter the rocks of a volcano flank, changing their appearance, composition and strength. Most commonly, hydrothermal alteration can weaken the rocks and promote failure or close permeable fluid paths and induce phreatic explosions (Heap and Violay, 2021). Thus, we attempted to identify areas of localised alteration, by visually inspecting the volcano flanks using Sentinel-2 data. We specifically looked for characteristic bright spots indicative of alteration minerals and localised vegetation loss (outside the main crater). If these were identified, we added 10 points.



- Topography between an edifice and the sea: Within 20 km of the shore, the volcano's flank may not directly face the sea. While past events at St. Helens, USA, 1980 (Fisher, 1990) and Merapi, Indonesia, 2010 (Cronin et al., 2013) have shown that sector collapses or PDCs can overcome significant topography, it is less likely to reach the sea in such circumstances. To account for that, we added 30 points if we found no major topographic obstacle between the summit of the edifice towards the sea, thus adding more points to volcanoes close to the coast.


For the chosen factor weights we decided to favour morphometry and eruptive activity over the others. Morphometry, here meaning H/D-ratio and slope angle, measure both the feasibility of gravitational mass movements (flank collapses or PDCs)
reaching the sea, as well as quantify oversteepening of individual flanks. Eruptive activity is also favoured as tsunamis do not only occur by flank collapse but also through PDCs or explosions. Additionally, unrest or eruptions may also act as a trigger for gravitational failures. In turn, we decided to weigh the Hazardous Features less since these are not quantitatively determined and more prone to human subjectivity and misjudgement. Consequently, the final factor weights used were the H/D-ratio (20%) and the slope angle (20%) as morphometry factors, then volcanic activity (30%), tsunamigenic history
(20%), and hazardous features (10%). An example how the score was calculated is provided in Fig. 2.



| c | H/D-Ratio | Volcanic Activity | Tsunamigenic History | Slope Angle | Hazardous Features | Total Score |
|---|---|---|---|---|---|---|
| Factor Value/ Feature | 771/1560 = 0.50 | Last erupted: 1968 | None are known | Max: 34° | - Caldera (15)<br>- Secondary vent (15)<br>- Mostly vegetation free slope (5)<br>- Hydrothermal Alteration (10)<br>- Submerged edifice (20)<br>- Unobstructed path to sea (30) | |
| Factor Points (Weight) | 50 (20%) | 77 (30%) | 0 (20%) | 67 (20%) | 95 (10%) | **56** |

**Figure 2:** Exemplary calculation of the ranking score using Nila volcano, Indonesia. a) is a false colour image from Sentinel-2 from bands 8 (NIR), 4 (red), and 3 (green). This highlights vegetated areas as red, but leaves barren areas grey. Here, Nila Island is densely vegetated, except for the southeast slope, which appears to be unvegetated due to ongoing hydrothermal alteration. b) is a hillshade DEM from Copernicus GLO30 and c) is a summary of our MCDA score calculation using data marked in a) and b) as well as the GVP and Global Historical Tsunami databases.

We further tested how robust our ranking is with respect to used factor weights. This is done to confirm that the highest scoring volcanoes still retain their high score even when the weighing is significantly different, which can confirm that these volcanoes really pose the highest tsunami hazard despite possible human error or misjudgement. The test was carried out by



changing the five factor weights, increasing one factor to 60% and all others are set to 10%. The procedure was repeated for all five factor weights, so that every single factor was once set as the strongest influence. We then counted how many times 250 the volcanoes would be placed within the top 5, 10, 15, and 20 spots on the ranking. For example, if a volcano is always in the top 5, no matter how the factor weights change, then it is 5 times in the top 5, 10, 15, and 20 each, giving it the maximum count of 20. However, if the volcano is once in the top 10, twice in the top 15 and four times in the top 20 it receives a count of 7. The resulting compilation was used to judge whether our ranking can generally identify the highest scoring and most hazardous volcanoes well.


| Volcano | Country | Year | Deaths | Cause | References |
|---------|---------|------|--------|-------|------------|
| Agung | Indonesia | 1963 | | | 3, 5 |
| Awu | Indonesia | 1856 | 3000 | Pyroclastic flows | 1, 2, 3, 4, 5 |
| Awu | Indonesia | 1892 | 1532 | Pyroclastic surges? | 1, 2, 3, 4, 5 |
| Awu | Indonesia | 1913 | | Pyroclastic surges? | 4 |
| Banua Wuhu | Indonesia | 1889 | | Underwater explosion? | 2, 3, 4 |
| Banua Wuhu | Indonesia | 1918 | | Underwater explosion | 2, 3, 4 |
| Banua Wuhu | Indonesia | 1919 | | Underwater explosion | 2, 3, 4 |
| Gamalama | Indonesia | 1608 | <50 | | 2, 3, 4, 5 |
| Gamalama | Indonesia | 1771 | | | 3 |
| Gamalama | Indonesia | 1772 | 35 | | 4 |
| Gamalama | Indonesia | 1840 | | | 2, 3, 4 |
| Gamalama | Indonesia | 1871 | | | 1 |
| Gamkonora | Indonesia | 1673 | | Earthquake/ Landslide | 5 |
| Gamkonora | Indonesia | 1673 | | | 2, 3, 5 |





| Iliwerung | Indonesia | 1973 | | Underwater explosions? | 2 |
|---|---|---|---|---|---|
| Iliwerung | Indonesia | 1979 | >539 | Landslide | 2, 3, 4 |
| Iliwerung | Indonesia | 1983 | | Underwater explosion | 2, 4, 5 |
| Krakatau | Indonesia | 416 | <1000 | | 3, 4, 5 |
| Krakatau | Indonesia | 1883 | 34417 | Pyroclastic flows | 1, 2, 3, 5 |
| Krakatau | Indonesia | 1883 | | Landslide? | 2, 5 |
| Krakatau | Indonesia | 1884 | | Underwater explosion? | 2, 3, 4 |
| Krakatau | Indonesia | 1928 | | Underwater explosion | 1, 2, 3, 4, 5 |
| Krakatau | Indonesia | 1930 | | Underwater explosion | 2, 3, 4, 5 |
| Krakatau | Indonesia | 1981 | | Landslide? | 2, 3 |
| Krakatau | Indonesia | 2018 | 437 | Landslide | 4 |
| Makian | Indonesia | 1550 | | | 2 |
| Peuet Sague | Indonesia | 1837 | | | 3 |
| Rokatenda/Paluweh | Indonesia | 1927 | 226 | Underwater explosion? | 1, 4 |
| Rokatenda/Paluweh | Indonesia | 1928 | 98 | Landslide | 1, 2, 3, 4 |
| Ruang | Indonesia | 1871 | >400 | Lava dome collapse | 2, 3, 5 |
| Ruang | Indonesia | 1889 | | | 1 |
| Soputan | Indonesia | 1845 | 118 | Earthquake? | 2, 3 |
| Tambora | Indonesia | 1815 | <1000 | Pyroclastic flows | 1, 2, 3, 4, 5 |
| Teon/Serawerna | Indonesia | 1659 | | Pyroclastic flows? | 2, 3, 4, 5 |
| Unknown Volcano | Indonesia | 1773 | | | 5 |



| Unknown Volcano | Indonesia | 1878 | | | 5 |
|---|---|---|---|---|---|
| Unknown Volcano | Indonesia | 1883 | | Pyroclastic flows? | 3 |
| Unknown Volcano | Indonesia | 1892 | | | 5 |
| Unknown Volcano | Indonesia | 1918 | | | 5 |
| Unknown Volcano | Indonesia | 1919 | | | 5 |
| Kadovar | Papua New Guinea | 2018 | | Lava dome collapse | 5 |
| Long Island | Papua New Guinea | 1660 | | Pyroclastic flows? | 2, 5 |
| Rabaul | Papua New Guinea | 1878 | | Earthquake | 2 |
| Rabaul | Papua New Guinea | 1937 | <50 | Pyroclastic flows/ Explosions | 2, 5 |
| Rabaul | Papua New Guinea | 1994 | | Pyroclastic flows | 2, 5 |
| Ritter Island | Papua New Guinea | 1888 | <3000 | Landslide | 2 |
| Ritter Island | Papua New Guinea | 1972 | | Underwater explosions? | 2, 5 |
| Ritter Island | Papua New Guinea | 1974 | | Landslide? | 2, 5 |
| Ritter Island | Papua New Guinea | 2007 | | Landslides? | 2 |
| Unknown Volcano | Papua New Guinea | 1857 | | Earthquake | 2 |
| Unknown Volcano | Papua New Guinea | 1953 | | | 5 |
| Bulusan? | Philippines | 1933 | 9 | | 2, 5 |
| Camiguin | Philippines | 1871 | | Pyroclastic flows? | 2, 5 |
| Didicas | Philippines | 1969 | 3 | Underwater explosions? | 2, 5 |
| Taal | Philippines | 1716 | | Underwater explosions | 2, 5 |
| Taal | Philippines | 1749 | | Pyroclatic flows? | 2, 5 |



| Taal | Philippines | 1754 | 12 | Pyroclatic flows | 2, 5 |
| Taal | Philippines | 1911 | >50 | Pyroclastic surges/ Air waves? | 2, 5 |
| Taal | Philippines | 1965 | 355 | Air waves? | 2, 5 |

**Table1:** Compiled list of historic Tsunami events in the Southeast Asia region. References are 1: Hamzah et al. (2000); 2: Paris et al. (2014); 3: Mutaqin et al. (2019); 4: Hidayat et al. (2020); 5: NGDC (2021).

### 2.3 Ranking assumptions and limitations

There are many factors influencing the tsunami hazard posed by volcanoes and not all of them can be quantitatively evaluated. Furthermore, even for the ones which can be evaluated, certain assumptions have to be made in order to create a comparable baseline to judge all volcanoes on an equal basis. Here we address some of the issues with the data used in our ranking approach:

It is important to emphasise that the point system used in our MCDA is based on arbitrary point scales, assigned to best cover the range of values used to build the ranking score. Any evaluation of this type will inherently involve arbitrarily chosen scores that have to consider and weigh the many multifaceted factors that contribute to volcano flank instability and the hazards of generating a tsunami. This also means that the factor points and weights are based on our subjective judgement by how important we think these factors are. Naturally, this leaves a lot of room for arguments on how the points and weights could be assigned differently, which may significantly change the final score. However, as the rules with which points are given are kept strictly the same for all volcanoes, we retain the comparability of our results, allowing a meaningful identification of the most hazardous volcanoes for tsunami generation. This is despite the individual score by itself having little meaning in terms of hard data, such as expected tsunami event frequency, possible wave heights, or impacts on shorelines and population. Similarly, we can thus not adequately assess the risk to shores and population in the traditional sense.

Another important consideration is that we cannot consider all factors that are known or suspected to impact the tsunamigenic potential of a volcano. The most noteworthy ones are (1) ongoing flank deformations (e.g., through dyke intrusions or slow décollement movement), which can destabilise parts of the edifice, and (2) absence of bathymetric data covering the underwater geometry of the volcanoes. Because the lack of high quality and accessible data for all volcanoes makes meaningful comparison of the impact of these factors on the tsunami hazard impossible, we could only consider these factors by proxy. For instance, eruptive activity of a volcano increases the likelihood of ongoing deformation and thus it is

 

indirectly tied to the activity factor points. However, not all erupting volcanoes are likely to experience significant deformation and, on the other hand, there may be significant deformation without an eruption (e.g., due to the intrusion of magma underground). For the underwater geometry, our analysis is simplified to whether the volcano is partially or fully in the water, thus counting towards the hazardous features factor points. But it neglects potential steep underwater flanks, which also means that some potentially hazardous submarine volcanoes had to be excluded from our catalogue as they had no subaerial edifice to examine. Here, these are Banua Wuhu, Indonesia, and Hankow Reef, Papua New Guinea. Future studies could take these missing factors into account properly and warrant their own point scores, provided that data quality, regularity, and availability of surveys improve.

Data based on historic observations is often flawed as they were not always systematically recorded. This includes historical tsunamis, flank collapses, and eruptions. The problem is facilitated the further the events lie back in time, and many events may be missed or wrongly interpreted because they were either not understood properly or simply not noticed or remembered. For instance, some volcanoes have unknown eruption dates and thus received a relatively low rating, even when it is quite likely that its last eruption happened only decades ago. One example is Balbi volcano, Papua New Guinea, which likely erupted around the early 1800s, but this is unconfirmed (Global Volcanism Program, 2013). This creates a slight bias towards volcanoes that are well-known and researched, whereas volcanoes with less scientific attention may receive lower ratings. Thus, our analyses may be prone to missing or incorrectly recorded events. For tsunamis, we tried to minimise this bias by incorporating data from multiple databases and studies (Hamzah et al., 2000; Paris et al., 2014; Mutaqin et al., 2019; Hidayat et al., 2020; NGDC, 2021). Surprisingly, no study had the exact same tsunami events listed and some were only found in certain reviews. Our compilation thus likely marks the most comprehensive list of historic tsunamis in SE-Asia to date, however, there are likely some further events that were missed or not recorded.

For evidence of edifice instability the problem is similar. Many collapse scars related to such instability or lateral blasts can be obscured by the regrowth of the volcano and are thus easy to miss. On the other hand, gradual erosion and destabilisation due to hydrothermally weakened rocks (Darmawan et al., 2022) may produce scars that are very similar to collapse scars and consequently very challenging to distinguish using satellite data. Finally, submarine debris avalanche deposits which are evidence for past edifice failures that reached the sea are poorly studied since the required bathymetry data is rarely acquired. Here, Silver et al. (2009) was specifically investigating volcanic debris avalanches for most volcanoes in Papua New Guinea, so this region can be considered reasonably well covered. However, no similar studies exist for Indonesia or the Philippines, making some oversights likely.

When analysing the vegetation cover of the volcanoes using the Sentinel-2 satellite images we found that the vegetation is subject to seasonal changes. Mainly this encompasses vegetation colour changes due to dry or rainy periods. Since our analysis is kept rather simple and qualitative rather than quantitative, this effect is unlikely to impact our results.




While we consider gravitational instability of volcanic edifices by measuring the slope steepness, additional factors can play an important role that could not be considered here as they are unknown for most volcanoes. These are mainly the lithological properties of the flank and the rock mass strength (Watters et al., 2000). Furthermore, the nature of the volcanic flanks deposits and their state of consolidation is likely to play a role on how resistant they are to oversteepening and destabilisation. For instance, one factor that is speculated to have contributed to the 2018 Anak Krakatau flank collapse is that it consisted of loose unconsolidated pyroclastics covered with lavas (Grilli et al., 2019), which likely contributed to the instability.

## 3 Results

### 3.1 Volcano catalogue and ranking

Using the factor points and weights described in section 2.2, we ranked the volcanoes in our catalogue by their tsunami hazard. An overview is presented in Fig. 3 and a list of the 40 highest-scoring volcanoes is shown in table 2. A complete, more detailed, and interactive version of this list with individual entries relating to how the points were counted can be found in the supplementary material. The points range from 76 (Batu Tara, Indonesia), representing the highest tsunami hazard, to 13 (Baluan, Papua New Guinea), representing the lowest tsunami hazard. Using these results, we grouped the volcanoes into high, medium, and low tsunami hazard categories based on their relative score. High hazard is assigned for volcanoes having more than 55 points (~14% of the volcanoes in the catalogue), medium hazard is assigned to volcanoes with scores between 40 and 55 points (~36% of the volcanoes in the catalogue) and low hazard category is assigned to volcanoes with less than 40 points (~49% of volcanoes in the catalogue). Volcanoes like Anak Krakatau, Indonesia, and Ritter Island, Papua New Guinea are among the highest tsunami hazard volcanoes in our ranking, which is little surprise considering both their history of powerful eruptions and catastrophic tsunamis (Paris et al., 2014). However, we also identify high high-hazard volcanoes that are not as prominently considered for their tsunamigenic potential, but received similarly high scores. In Indonesia these include Batu Tara, Gamalama, Iliwerung, Karangetang, Nila, Sangeang Api, Wetar, Sirung, Serua and Ruang. For Papua New Guinea we identify Kadovar, Rabaul, Ritter Island, Manam, Bam, Langila and Ulawun as volcanoes with high tsunami hazard. For the Philippines, only Didicas is classified as a high tsunami hazard volcano.





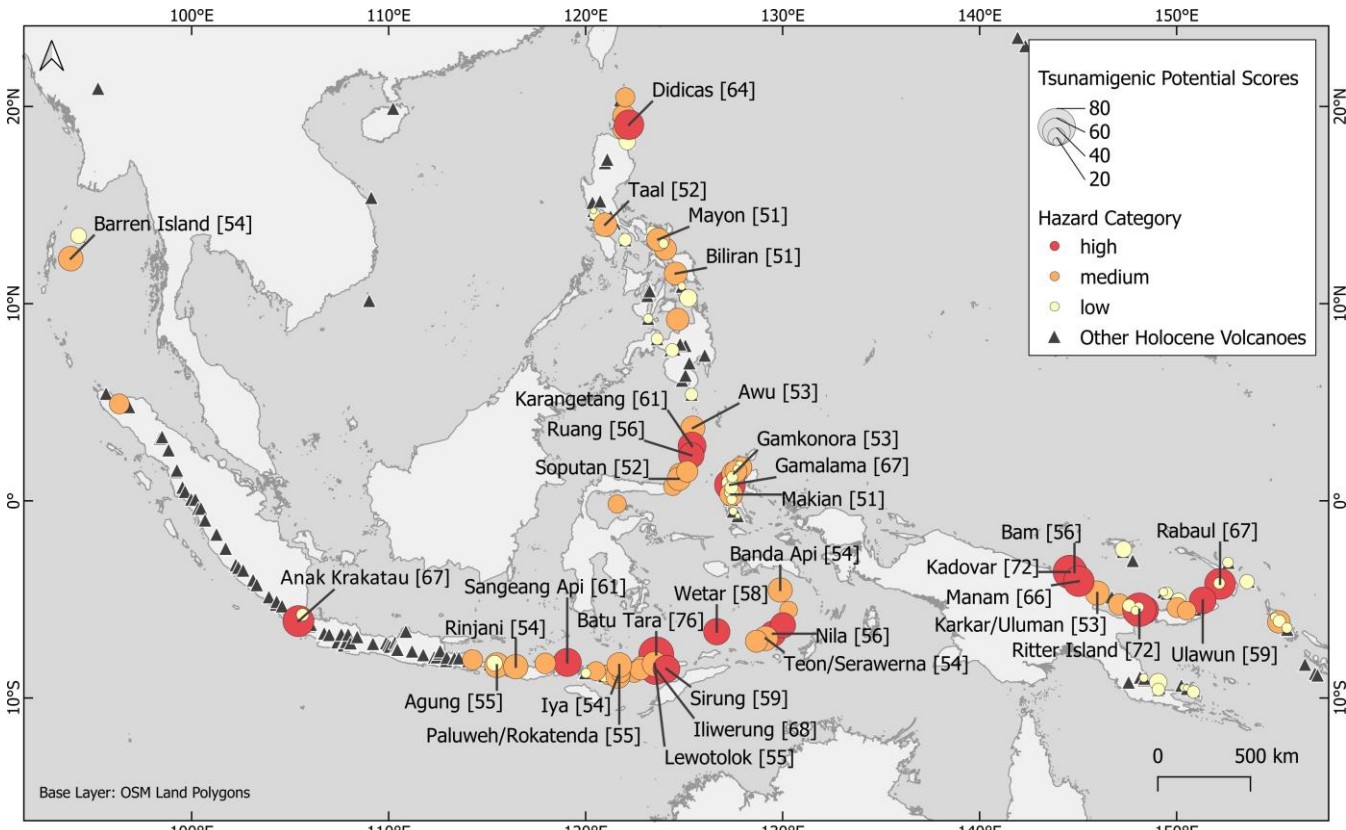

**Figure 3:** Maps of the volcanoes in the catalogue with their corresponding ranking score and the resulting hazard category. For a detailed overview of tsunami hazard scores for all volcanoes in the catalogue, see table 2. Base map data source: © OpenStreetMap contributors 2022. Distributed under the Open Data Commons Open Database License (ODbL) v1.0.

| Volcano | Country | Points: H/D-Ratio | Points: Volcanic Activity | Points: Tsunamigenic History | Points: Slope Angle | Points: Hazardous Features | Total weighted Score |
|---|---|---|---|---|---|---|---|
| Batu Tara | Indonesia | 83 | 95 | 20 | 100 | 65 | **76** |
| Anak Krakatau (pre-2018) | Indonesia | 52 | 100 | 80 | 60 | 80 | **76** |
| Kadovar | Papua New Guinea | 79 | 100 | 20 | 74 | 80 | **72** |
| Ritter Island | Papua New Guinea | 100 | 87 | 60 | 39 | 65 | **72** |
| Iliwerung | Indonesia | 45 | 100 | 40 | 70 | 70 | **68** |
| Anak | Indonesia | 21 | 100 | 90 | 36 | 80 | **67** |





Krakatau

| | | | | | | | |
|---|---|---|---|---|---|---|---|
| Gamalama | Indonesia | 40 | 98 | 50 | 63 | 70 | **67** |
| Rabaul | Papua New Guinea | 62 | 94 | 30 | 67 | 70 | **67** |
| Manam | Papua New Guinea | 41 | 100 | 20 | 76 | 90 | **66** |
| Didicas | Philippines | 89 | 78 | 10 | 73 | 65 | **64** |
| Sangeang Api | Indonesia | 36 | 100 | 20 | 67 | 70 | **61** |
| Karangetang | Indonesia | 49 | 100 | 0 | 72 | 70 | **61** |
| Langila | Papua New Guinea | 29 | 100 | 20 | 65 | 75 | **60** |
| Sirung | Indonesia | 47 | 95 | 0 | 62 | 85 | **59** |
| Ulawun | Papua New Guinea | 25 | 100 | 20 | 68 | 60 | **59** |
| Wetar | Indonesia | 86 | 50 | 10 | 86 | 65 | **58** |
| Nila | Indonesia | 50 | 77 | 0 | 67 | 95 | **56** |
| Ruang | Indonesia | 45 | 82 | 20 | 64 | 55 | **56** |
| Bam | Papua New Guinea | 50 | 76 | 20 | 68 | 55 | **56** |
| Serua/ Legatala | Indonesia | 70 | 73 | 0 | 67 | 65 | **56** |
| Lewotolok | Indonesia | 40 | 100 | 0 | 56 | 55 | **55** |
| Agung | Indonesia | 25 | 99 | 10 | 65 | 50 | **55** |
| Paluweh/ Rokatenda | Indonesia | 32 | 93 | 20 | 49 | 65 | **55** |
| Barren Island | India | 33 | 100 | 0 | 48 | 80 | **54** |
| Teon/ Serawerna | Indonesia | 51 | 71 | 20 | 63 | 60 | **54** |
| Rinjani | Indonesia | 20 | 96 | 0 | 67 | 75 | **54** |
| Iya | Indonesia | 49 | 77 | 10 | 64 | 60 | **54** |
| Banda Api | Indonesia | 52 | 79 | 0 | 62 | 70 | **54** |
| Gamkonora | Indonesia | 33 | 87 | 20 | 53 | 60 | **53** |

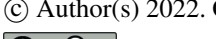


| | | | | | | |
|---|---|---|---|---|---|---|
| Karkar/ Uluman | Papua New Guinea | 26 | 94 | 10 | 49 | 80 | **53** |
| Awu | Indonesia | 24 | 84 | 40 | 48 | 55 | **53** |
| Taal | Philippines | 2 | 100 | 50 | 21 | 75 | **52** |
| Soputan | Indonesia | 10 | 100 | 20 | 53 | 50 | **52** |
| Mayon | Philippines | 25 | 99 | 0 | 64 | 35 | **51** |
| Makian | Indonesia | 36 | 79 | 10 | 57 | 65 | **51** |
| Biliran | Philippines | 32 | 74 | 20 | 70 | 40 | **51** |
| Camiguin | Philippines | 30 | 76 | 10 | 63 | 65 | **50** |
| Bagana | Papua New Guinea | 12 | 100 | 0 | 70 | 35 | **50** |
| Babuyan Claro | Philippines | 32 | 73 | 20 | 43 | 90 | **50** |

**Table 2:** List of the 40 highest scoring volcanoes in our catalogue and their respective MCDA ranking of the relative
tsunami hazard, showing individual factor points and the final score. The factor weights for the total score are H/D-ratio
(20%), volcanic activity (30%), tsunamigenic history (20%), slope angle (20%), and hazardous features (10%). For the full
catalogue and ranking table see the supplementary material.

The results of the ranking robustness testing are summarised in Fig. 4, showing that the higher scoring volcanoes in our
ranking generally still score high and are independent of the factor weight, which adds confidence to our results. The only
notable exceptions here are at the transitions between low and medium hazard categories (e.g., Hiri, Indonesia, or Tolokiwa,
Papua New Guinea) in Fig 4a and 4b, which is indicating that these volcanoes are likely more sensitive to individual weights
and, thus, less robustly ranked. The lower the volcanoes are ranked, the less robust the ranking order becomes, which is
likely due to a higher number of volcanoes with similar scores.






**Figure 4:** Robustness test of the factor weights used in the ranking. This counts the number of times a volcano made it into the top 5, 10, 15 and 20 places in the ranking with different individual weights. This is displayed per countries a) Indonesia, b) Papua New Guinea and c) Philippines. It generally shows that the volcanoes we classed as high, medium and low hazard are well generally sorted in the order we classed them in, despite different factor weights and with only few exceptions. This




shows that changing the factor weights may slightly change the order in which the volcanoes are ranked, but our analysis is generally classifying higher hazard volcanoes correctly, confirming the robustness of our ranking.

## 3.2 Volcano distribution and tsunami causes

Most of the high and medium tsunami hazard volcanoes are located in Indonesia, which by itself is not surprising since Indonesia also has the most volcanoes in our catalogue (~46%). However, the relative amount of these volcanoes in those categories is significantly higher (Fig. 5), suggesting that Indonesia has an over-proportionally high number of hazardous volcanoes. This is further evident in the low tsunami hazard category, which are dominantly from Papua New Guinea (Fig. 5). Volcanoes of the Philippines are only underrepresented in the high tsunami hazard category, but this may be due to the

lower number of overall volcanoes.

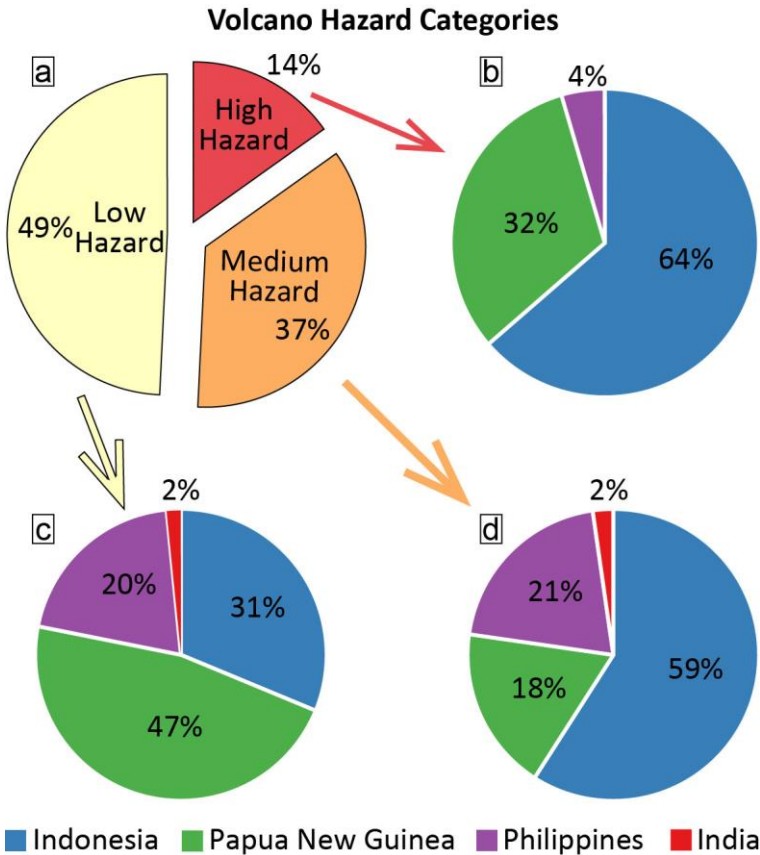

**Figure 5:** Tsunamigenic hazard from individual volcanoes by country. Shown are (a) all considered volcanoes and their hazard categories according to our ranking, then their respective distribution between the countries for (b) high hazard, (c)

medium hazard, and (d) low hazard volcanoes. We find a disproportionately large number of high and medium hazard





category volcanoes located in Indonesia (~64% and ~59%, respectively), which is much larger than expected since Indonesian volcanoes make up only ~46% of all volcanoes in our catalogue. Contrary, the Philippines have a lower share of high-hazard volcanoes (~4% only one volcano), but make up 18% of all volcanoes in our catalogues. Papua New Guinean volcanoes make up ~35% of the catalogue, so they can be considered underrepresented in the medium hazard category and

overrepresented in the low hazard category.

Our review of historical tsunamis in Southeast Asia contains 61 distinct events (Table 1) and shows that the majority of the historical volcanogenic tsunamis still have an unknown or uncertain cause (Fig. 6). However, we can still extract that the most known or suspected causes were explosions (21%), followed by mass movements like pyroclastic flows (19%) and

landslides (14%). Volcanic earthquakes rarely cause tsunamis by themselves since volcanic quakes are usually much weaker compared to tectonic events and lava dome growth is a rather specific eruption style and thus less frequent in producing tsunamis (here only 2 cases). Some cascading events also occurred involving multiple causes (here 3 cases), which were once an earthquake and a landslide, and twice an explosion and pyroclastic flow. It is noteworthy that most known cases of volcanogenic tsunamis are produced by pyroclastic flows and explosions, both of which are commonly associated with

strong eruptive activity. Gravitational failures such as landslides or lava dome collapses occur less frequently, but do not necessarily require ongoing eruptions.

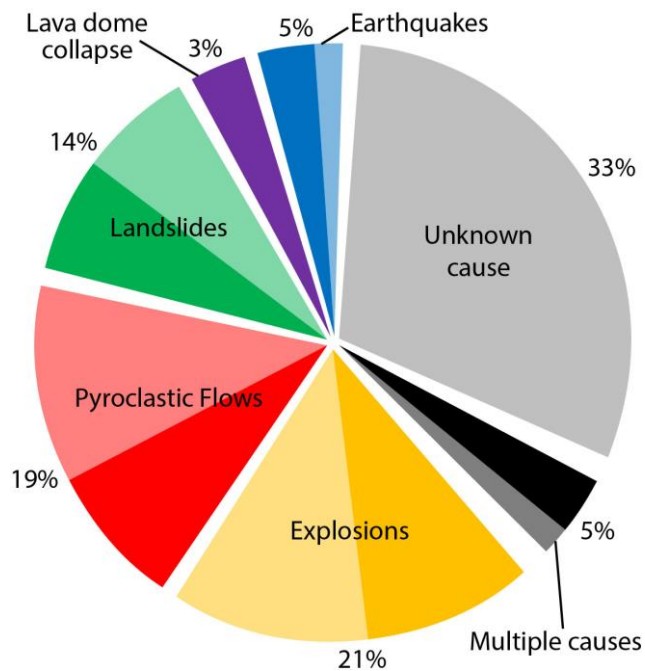

**Figure 6:** The historical causes of volcanogenic tsunamis in SE-Asia sourced from table 1, uncertain causes are marked

transparent. A third of all cases is caused by mass movements (pyroclastic flows and landslides) and about a fifth is



attributed to explosions, whereas earthquakes and lava domes rarely produce tsunamis. For the purpose of this figure, air waves were grouped with explosions as they also require a strong explosion.

## 4 Discussion

### 4.1 Evaluating scores and ranking through recent tsunami events

In recent years, both Anak Krakatau, Indonesia, and Kadovar, Papua New Guinea, produced tsunamis, with good-quality topographic data available and their circumstances being well known (Plank et al., 2019; Walter et al., 2019). One important aspect when judging the usefulness of our ranking is its ability to correctly identify volcanoes that are most likely to produce tsunamis in the future. We tested this by comparing the scores of both volcanoes as it is now compared to how it was before the event. The morphology of Anak Krakatau has changed quite significantly following the 2018 flank collapse (Darmawan

et al., 2020). The island is reduced in size and became lower in elevation, which consequently reduced the H/D-ratio factor points. However, it now has a crater that is open to the sea and it still has a history of many tsunamis and regular recent eruptions, thus only changing the score slightly from 76 points before the collapse to 67 points after (Table 2). Consequently, Anak Krakatau was the highest hazard volcano before its collapse and tsunami occurred, confirming that our approach would have correctly identified the volcano as a threat. Now, after the collapse, it is still among the highest scoring volcanoes,

meaning the recent 2018 tsunami and the event changed little in its overall status. However, the 2018 flank collapse indeed lowered the score, which is reasonable since the main volcanic edifice has yet to rebuild after the collapse. Further tsunamis through further collapses, explosions or pyroclastic flows are still potential tsunami causes that may occur at Anak Krakatau in the near future.

For Kadovar, the changes were different. While the 2018 eruption produced a new vent and formed a littoral lava dome

(Plank et al., 2019), the overall topography of the island did not change significantly. Here, the tsunami was not generated as a result of a major flank collapse but rather due to a collapse of the littoral dome and smaller parts of the southern flank (Plank et al., 2019). However, before the eruption, Kadovar had no known historic eruptions, although it is possible that one occurred in 1700, but this is unconfirmed (Llanes et al., 2009; Global Volcanism Program, 2013). In 1976 the island residents were briefly evacuated due to strong fumarolic activity and fears of an eruption, which ultimately did not occur

(Llanes et al., 2009). Thus, before its 2018 eruption, the island could only be considered to have historic unrest, which lowered the score significantly. The score was 45 points before the 2018 tsunami, meaning Kadovar would still have been identified as a volcano with an elevated hazard (medium category) before the eruption. However, given its high eruptive activity now, the score has increased strongly. If the unconfirmed eruption in 1700 is included, the hazard score would be significantly higher at 54 points, just outside the high hazard category. This highlights that a better constrained volcanic

history on some volcanoes can significantly improve the meaningfulness of this ranking. Now, after the 2018 tsunami, Kadovar scores 72 points, making it the third highest tsunami hazard in SE-Asia based on our ranking. Both the steep south-





facing flank and the littoral lava dome are still in place and may pose a large tsunami hazard in the future, especially if the eruptive activity continues.

From both these cases we can conclude that our ranking system is able to identify hazardous volcanoes reasonably well before a tsunami occurs. While individual volcano scores may change significantly over time due to eruptions or morphological changes, the applied multi-categorized approach has worked well for both Anak Krakatau and Kadovar. However, these cases also emphasise that particular attention should be devoted to coastal volcanoes with unclear eruptive histories, especially when they become active after decades or centuries of no activity.

## 4.2 Future tsunami hazards in SE-Asia

To allow for a more detailed look at future tsunami hazards in SE-Asia we summarised in which locations a high concentration of hazardous volcanoes is located. This was done by performing a weighted point density calculation, highlighting areas where many tsunamigenic volcanoes are located closely together, with their impact multiplied by the hazards score (Fig. 7). The result shows that the area with the highest volcanogenic tsunami hazard is located around the Indonesian Lesser Sunda Islands, particularly between East Nusa Tenggara and the Alor archipelago, at the Molucca Sea coast between northern Sulawesi and Halmahera, and at the southern Bismarck Sea in Papua New Guinea. Further elevated hazard areas can be found within the Indonesian Banda Sea, the Philippine Luzon Strait, the central Philippine Islands, and along the southern Solomon Sea coast of Papua New Guinea. These areas can thus be considered to be the most important to prioritise for tsunami monitoring, modelling and forecasting.



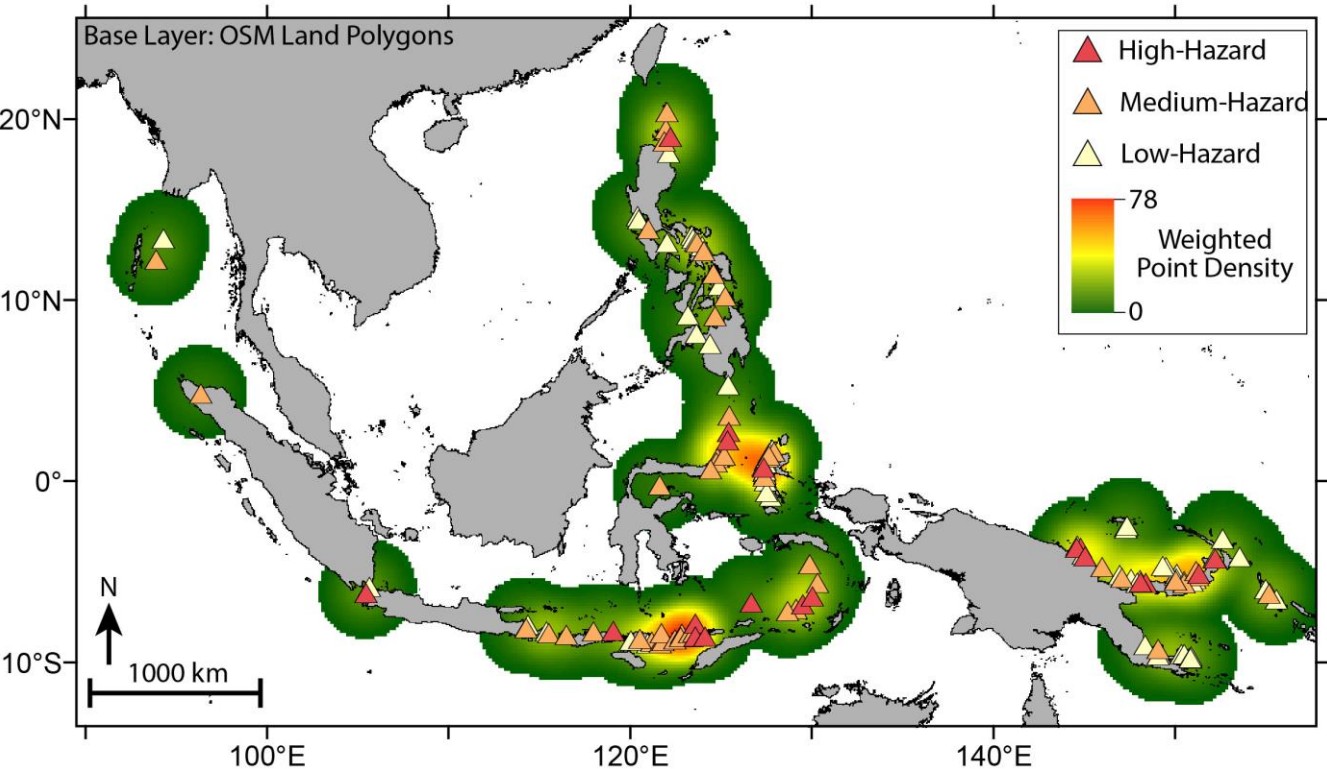

**Figure 7:** Heat map resulting from weighted point density calculation using the hazard score. Shown are the most likely source areas of volcanogenic tsunamis, with a higher density meaning the area is closer to many tsunamigenic volcanoes. This does not represent the travel distance or wave heights of a potential tsunami, which may affect more distal coasts, but only highlights the most likely source regions of a tsunami. Base map data source: © OpenStreetMap contributors 2022. Distributed under the Open Data Commons Open Database License (ODbL) v1.0.

To account for the potential spatial impact of volcanogenic tsunamis, we extend our tsunami hazard evaluation by assessing the total length of a coastline affected within one and two hours of tsunami propagation for the volcanoes categorised as high hazard in our ranking (except Didicas). For that, we compute tsunami travel times (TTT) from point sources centred at each volcano position using the SRM30+ bathymetry (Becker et al., 2009) resampled to 1 arc minute resolution and the numerical algorithm as proposed by Marchuk (2008). Fig. 8 shows tsunami propagation fronts after 1 hour of wave propagation. Table 3, in turn, lists the total lengths of the coastlines affected after 1 and 2 hours of propagation, respectively. Detailed tsunami travel time plots for the individual volcanoes can be found in the supplementary figures 1, 2 and 3. It is important to note that this type of simulation only includes the travel time of the tsunami, but not other important characteristics such as wave amplitude.



| Volcano | Coastline 1h in km | Coastline 2h in km |
|---|---|---|
| Anak Krakatau | 1621 | 3969 |
| Batu Tara | 7322 | 25511 |
| Kadovar | 3078 | 10195 |
| Ritter Island | 3933 | 13370 |
| Gamalama | 6069 | 27900 |
| Rabaul | 5007 | 16380 |
| Manam | 2720 | 9061 |
| Iliwerung | 5974 | 21244 |
| Sangeang Api | 5451 | 18112 |
| Karangetang | 5819 | 24147 |
| Langila | 3955 | 13174 |
| Sirung | 5866 | 22289 |
| Ulawun | 2521 | 11527 |
| Wetar | 8512 | 27519 |
| Nila | 6117 | 23948 |
| Ruang | 6019 | 24998 |
| Bam | 2841 | 9012 |
| Serua | 6498 | 23998 |

**Table 3:** The affected coastline of a tsunami originating from high hazard volcanoes extracted using the tsunami travel time modelling. We note that Anak Krakatau, where the most prominent event in recent years occurred in 2018, affects the lowest length of coastline. This highlights that similar events at other locations can have much more widespread impacts.

The results show the potentially wide reach of volcanogenic tsunamis in both Indonesia and Papua New Guinea (Fig. 8), but it also reveals surprising differences. Anak Krakatau is only projected to affect ~1600 km of coastline within one hour (Table 1) and the area is restricted mostly within the Sunda Strait coast and parts of southern Java and Sumatra. This is nearly





consistent with the 2018 tsunami, where the impacts were only within the Sunda Strait (e.g., Paris et al., 2020). All other volcanoes may produce tsunamis affecting a lot more coasts (Table 3). Batu Tara, which is now the highest scoring volcano

in our catalogue, is projected to affect a coastline of ~7300 km, almost 5 times as much as Anak Krakatau within the same time. The longest total affected coastline of ~8500 km (after 1 hour of propagation) is calculated for Wetar, covering almost the entire Banda Sea. Tsunamis from the northern volcanoes in Indonesia are shown to potentially reach up to the Philippines as well as cover large parts of northern Sulawesi and Halmahera and almost all volcanoes in Papua New Guinea (with the exception of Rabaul) reach through most of the Bismarck Sea.

Despite demonstrating the potential to affect large coastal areas, our modelling does not inform how severe a tsunami from a volcanic source could actually turn out. In fact, tsunami travel time modelling neither accounts for source magnitude, nor for energy transfer. Full source process simulation coupled to modelling of the full wave propagation is needed to assess the magnitude of the coastal impact. In particular, the models are expected to strongly depend on the mechanism triggering the tsunami, i.e., a landslide, PDC or explosion, as well as the magnitude of the event. For example, in landslide or sector

collapse events, the largest runups and the most severe impacts are expected to be largest in the near-field of the volcano, but may still be significant in the far-field (Harris et al., 2012; Grilli et al., 2021). Furthermore, the recent Hunga Tonga-Hunga Haʻapai eruption has shown that large eruptions are capable of generating meteotsunamis travelling long distances without significant loss of amplitude. This particular tsunami also travelled faster than initially expected (Somerville et al., 2022). To compensate for these knowledge gaps, full physics-based source and propagation modelling, especially if they are coupled to

the population density information along the coasts, may become the most useful tool to improve our understanding of the risk posed by volcanogenic tsunamis. So far, published models almost exclusively consider Anak Krakatau (e.g., Grilli et al., 2019; Heidarzadeh et al., 2020; Mulia et al., 2020; Omira and Ramalho, 2020; Paris et al., 2020), whereas the risk posed by the other high-hazard volcanoes in our catalogue is largely unknown, emphasising the need for future investigations and modelling efforts.




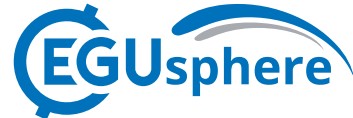

**Figure 8:** Travel-distance plots of tsunamis originating at a high-hazard volcano after 60 minutes in a) Indonesia and b) Papua New Guinea. How much coastline is affected by each volcano is summarised in table 3. Base map data source: © OpenStreetMap contributors 2022. Distributed under the Open Data Commons Open Database License (ODbL) v1.0.






### 4.3 Tsunami source volcanoes

Looking at individual volcanoes, our catalogue and ranking have identified a large number of potentially hazardous ones regarding the production of tsunamis. In the high hazard category are the Indonesian volcanoes Anak Krakatau, Batu Tara, Gamalama, Iliwerung, Sangeang Api, Karangetang, Sirung, Wetar, Nila, Ruang and Serua. In Papua New Guinea high risk

volcanoes include Kadovar, Ritter Island, Rabaul, Manam, Langila, Ulawun and Bam. In the Philippines there is only one high risk volcano - Didicas. Below, we briefly elaborate on the aspects that contribute to the high score of these volcanoes, speculate on the nature of future tsunamis that can be expected from them, and assess which particular volcanoes should be prioritised for tsunami monitoring and forecasting. Anak Krakatau and Kadovar have been discussed in the previous section.

**Batu Tara:** This lone and small volcanic island is located north of the Lesser Sunda Islands and positioned centrally

between the Flores- and Banda Sea. The island is exceptionally steep with the eastern flank measuring an incline of about 50 degrees. The morphology here is strikingly reminiscent of the Sciara del Fuoco at Stromboli, Italy. Despite its relatively small size, the Island also has an elevation of 753 m, making it 2.5 times higher than Anak Krakatau before its collapse. While no historical tsunamis are known from this volcano, the dissected morphology and apparent amphitheatre remnants suggest that multiple collapses have occurred during Batu Tara's geological history. It is also a recently active Island with the

last activity being in 2015, consisting mostly of strombolian and vulcanian eruptions, but also pyroclastic flows and rockfalls were reported to reach the sea (Global Volcanism Program, 2013). All these factors suggest that this volcano is a particularly large tsunami hazard, and the tsunamis could be generated both by catastrophic and smaller sector collapses due to gravitational instability of the oversteepened flanks as well as eruptive activity with large explosions and pyroclastic flows, which have a direct and steep path towards the sea. Travel-time modelling of a potential tsunami at this location is estimated

to affect all lesser Sunda Islands, most of the Banda Sea, the north of Timor as well as the southern parts of Sulawesi and Buru within an hour (Fig. 8). Since this volcano is a lot less known compared to islands like Anak Krakatau, *future monitoring efforts and tsunami risk modelling should make Batu Tara a high priority target*.

**Iliwerung:** The small cone is situated at the southern coast of Lembata Island, where it forms a complex of vents and lava domes, including some submarine ones. The main subaerial cone is located above a steep flank, providing a direct path

towards the sea. The volcano is also known for frequent eruptions with the last one being submarine in 2021 (Global Volcanism Program, 2013), and has three recorded historical tsunamis in 1973, 1979, and 1983 (Table 1), as well as signs of a past sector collapse in its morphology. Considering all above, Iliwerung is one of the highest ranking volcanoes in our catalogue. All known mechanisms of volcanogenic tsunami generation may be relevant here, especially large submarine or coastal explosions, pyroclastic flows and flank or lava dome collapses. A potential tsunami would likely affect all Lesser

Sunda Islands and well as Timor in a short amount of time (Fig. 8).



**Gamalama:** The volcano forms a large, nearly circular island in the north of Indonesia as part of the Maluku Islands at the Molucca Sea. It has the city Ternate on its eastern flank, which has approximately 205,000 inhabitants, making it the largest and most densely populated city in the province and an important economic centre. Since the volcano is very frequently active with its last eruption in 2018, it has both a history of deadly eruptions such as in 1775 and 2011 (Hidayat et al., 2020)
as well as tsunamis in 1608, 1771, 1772 and 1840 (Table 1). Since most of the volcano's flanks are not exceptionally steep, the most likely causes for tsunamis are far-reaching pyroclastic flows or eruptions on the lower flanks, however, partial collapses from the steeper summit region should also be considered. While Ternate is the largest populated area affected, it should be noted that a potential tsunami could also affect the other nearby islands as well as the western coast of Halmahera, the eastern coast of North-Sulawesi as well as the Islands Taliabu and Mangole.

**Sirung:** Similar to Iliwerung, Sirung forms a peninsula towards the south of the Lesser Sunda Islands. It is located on Pantar Island and forms a multifaceted complex including a large caldera, a steep stratocone and lava domes. Recent eruptions have been dominantly phreatic, although the morphology suggests that multiple eruptive styles are possible, including potential large caldera-forming eruptions. While these are less likely to occur, the resultant pyroclastic flows would need to travel less than 3 km downhill to reach the sea. As with Iliwerung, a potential tsunami would rapidly affect all Lesser Sunda Islands and
the north coast of Timor.

**Ritter Island:** After its catastrophic sector collapse and tsunami in 1888 much of the island's subaerial edifice remains destroyed, leaving only an elongated ridge as a remnant scar. While this may exaggerate the morphological metrics applied to our ranking (the island largely consists of a west-facing scar), the volcano has still produced multiple tsunamis after the large collapse, which occurred in 1972, 1974 and 2007. This demonstrates the volcano's continued potential to produce
tsunamis, both by explosions and sector failures of the subaerial or submarine edifice, but also a scenario similar to the recent Hunga Tonga-Hunga Haʻapai eruption (Somerville et al., 2022) should be considered since both these volcanoes have their main vents located in shallow waters.

**Rabaul:** The large 8 by 14 km Rabaul caldera forms a bay south of the Gazelle Peninsula in the northeast of New Britain. Here, it is questionable how well our morphological criteria for the ranking represent the tsunami hazard as the subaerial
edifice is limited to the Tavurvur and Vulcan cones, which are on opposite ends of the submerged caldera. We chose Tavurvur since it was the site of Rabauls most recent eruption in 2014. On the other hand, the volcano's tsunamigenic potential has been demonstrated in 1878, 1937 and 1994, where all tsunamis occurred in conjunction with significant explosive eruptions. Since the caldera is mostly submerged, tsunamis may be generated not only by pyroclastic flows and edifice collapses at the two main cones, but also underwater or coastal explosions. Even otherwise less significant phreatic
eruptions around the hot springs adjacent to Tavurvur may be considered here. A tsunami would affect the city of Rabaul directly inside the bay, but may also spread to the northern coast of New Britain, the western coast of New Ireland and the Duke of York Island.



**Didicas:** Lava dome extrusion from volcano formed a small island in 1952 in the Luzon Strait in the north of the Philippines. Multiple repeated eruptions are known from this volcano since 1773, which were mostly submarine, although some islands
were formed and destroyed prior to the current island. The last eruptive episode ended in 1978, but had produced a tsunami in 1969. Due to the small size of the young island, future activity has the potential to cause tsunamis by full or partial failure of the edifice, underwater explosions and lava dome collapses, should further domes grow. A tsunami from this location is likely to affect the Babuyan and Batanes Islands as well as the northern coast of Luzon. We note that Didicas was not used for the tsunami travel time simulations.

**Karangetang, Sangeang Api and Manam:** These three volcanoes all form larger, near circular volcanic islands (diameters between 8-15 km), with Karangetang connecting to the southern part of Siau Island. They are among the most active volcanoes on this list, having regular and dominantly explosive eruptions of varying intensity. Currently, Karangetang and Manam have ongoing eruptions at the time of writing and Sangeang Api had its last in 2020. While the lower flanks are mostly forested with gentle slopes, the summit regions are very steep and barren due to the constant activity. Manam and
Sangeang Api are also heavily dissected, suggesting multiple partial edifice failures have occurred in their geological history. Despite this, no historical tsunamis are known from any of these volcanoes, but gravitational mass movements like pyroclastic flows from large explosions or lava dome collapses as well as landslides from edifice failures have direct downhill paths into the sea in multiple directions, but would need to travel between 3 and 7 km. This makes a scenario in which a tsunami is generated only likely for larger eruptions. But as Anak Krakatau demonstrated, significant sector failures
may also occur without significant eruptions (Williams et al., 2019).

**Ruang, Serua, Nila, Wetar and Bam:** The volcanoes grouped here are the smaller volcanic Islands (diameters under 5 km). Naturally, these are primed for tsunami generation as their flanks are small and steep, with eruptions occurring close to the sea, however, most of these volcanoes are not as frequently active compared to the larger islands Karangetang, Sangeang Api and Manam. Their last eruption ranges from decades (Ruang, Nila, Wetar, Bam) to a century ago (Serua) and - with the
exception of Ruang in 1871 and 1889 - none of them have associated historical tsunamis. On the other hand, all islands show signs of past edifice collapses, which is confirmed for Bam through submarine debris avalanche deposits (Silver et al., 2009), which underlines their tsunamigenic potential. Ongoing hydrothermal alteration is also visible on the flanks of Wetar, Nila and Serua, potentially weakening their flanks. For the consideration of future tsunami hazards posed by these volcanoes, especially since these smaller islands with little to no habitation are less studied, *it is important to have adequate monitoring*
*in place as renewed eruptive activity would make these volcanoes particularly likely tsunami sources*. Partial flank failures similar to Anak Krakatau in 2018, both subaerial and submarine are likely causes, but explosions and pyroclastic flows are also possible.

**Langila and Ulawun:** The two Papua New Guinean volcanoes Langila and Ulawun are large and steep stratovolcanoes, very similar to large volcanic Islands such as Karangetang, both in terms of morphology and activity (frequent explosive





eruptions in recent years) as well as no known historical tsunamis. One major difference is that both are located on land and have only parts of their flanks facing the sea, west to northeast for Langila and only northwest for Ulawun. This also makes coastal flank eruptions less likely and the main tsunami hazard stems from far-reaching pyroclastic flows (up to 9.5 km for Ulawun) or major edifice failures (both volcanoes have signs of past collapses).

**Other relevant volcanoes:** This final paragraph briefly highlights some volcanoes that were not classified into the high
hazard category, but should nonetheless be considered for tsunami assessments. The Indonesian volcanoes Awu and Paluweh (or Rokatenda) and the Philippine volcano Taal do not have as tall and steep edifices as some other volcanoes on this list and thus received a lower score. However, all have a history of producing tsunamis with hundreds to thousands of fatalities as a result of their eruptions (Table 1). Provided that eruptions resume it is likely that such a scenario can happen again in the future. The latest eruption at Paluweh occurred between October 2012 and August 2013 in which an effusive-
explosive eruption produced PDCs that caused 5 fatalities, however, fortunately the pyroclastic materials did not trigger tsunami (Primulyana et al., 2017). An eruption at Taal is currently ongoing at the time of writing, highlighting the relevance of these volcanoes. Similarly, there are a number of small volcanic islands that did not score as high because they are not as active, meaning their last eruption occurred decades to centuries ago. For Indonesia, these are Teon, Manuk, Wurlali, and Banda Api. Here, the situation is comparable to Kadovar before it erupted in 2018, the islands could become a significant
tsunami hazard if new eruptive activity resumes.

## 5 Conclusions

Based on our MCDA analysis considering 131 volcanoes in SE-Asia we identify 19 that pose a high tsunami hazard and another 48 with moderate tsunami hazard. We find that the Indonesian Lesser Sunda Islands and northern Molucca Sea as well as the southern Bismarck Sea in Papua New Guinea are areas with a high number of hazardous volcanoes and may thus
be particularly prone to tsunamis sources by volcanoes. Many of these volcanoes such as Batu Tara, Indonesia, are not commonly considered for this type of hazard. We therefore emphasise the need to reconsider the current state of monitoring and risk assessment in these areas. Since tsunami warning systems are mostly not designed to detect volcanogenic tsunamis, our results highlight the importance of a reassessment of the current network and additional suitable equipment on the ground and through earth observation satellites. Due to the inherently short warning times of these events, we also
recommended increased pre-emptive measures on a local level, such as increased public education programs for coastal communities and the marking evacuation routes along populated coasts.



# 6 Acknowledgements

The authors acknowledge the financial support by the Federal Ministry of Education and Research of Germany in the framework of the TSUNAMI_RISK project (project numbers 03G0906A and 03G0906B), which is a part of the funding
initiative CLIENT-II. We further acknowledge the past contributions by GITEWS, on which this projects builds upon

# 7 Author contributions

EZ conceptualised and wrote the manuscript. Figures were prepared by EZ and AO. EZ also performed the analyses and evaluations of the individual volcanoes. EZ, AO, SP, TW and IR jointly developed the point score system. Tsunami-travel-time modelling was performed by AB and photogrammetric data for Anak Krakatau was provided by HD. All authors
contributed to the writing and editing of the manuscript.

# 8 Competing interests

The authors declare no competing interests.

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
