# Peer review of "Identification and ranking of subaerial volcanic tsunami hazard sources in Southeast Asia"

_EGUsphere, 2022_

## Author Comment (AC3)

The manuscript entitled "Identification and ranking of volcanic tsunami hazard sources in Southeast Asia" by Zorn et al. proposed a catalogue of potentially tsunamigenic volcanos in Southeast Asia and ranked these volcanoes by their tsunami hazards. The evaluation is based on a Multicriteria Decision Analysis (MDA) composed of five weighted factors. They identified 19 volcanoes with high tsunami hazard and 48 with moderate tsunami hazard. The proposed ranking system can identify the hazards of Anak Krakatau and Kadovar before a tsunami occurs as a retroactive study.

I agree that this study is meaningful to disaster mitigation of volcanic tsunamis. However, the ranking system proposed in this study is not objective and in lack of quantitively evidence to support the assessment. Meanwhile, the linear combination of five individually weighted factors for MDA are questionable. Unfortunately, the present form is not suitable for publication in Natural Hazards and Earth System Sciences. Significant additional work is required to improve the methodology and contents. My suggestion is that the manuscript should be revised substantially and resubmitted. I am willing to review this manuscript again after their revision. Here are my comments on this manuscript.

**Reply: We appreciate the feedback and the constructive comments that helped us to improve and clarify many points related to ranking system, objectivity of the analysis and interpretation. We especially improve the methods related to the individually weighted factors for the MCDA, we quantify the uncertainties, and rewrite large parts of the discussion section, where limitation aspects are now much more carefully illuminated. By responding to this point and the more specific points below, our manuscript has much improved.**

1. My first concern on this manuscript is that the ranking system is not objective. The scoring (F) is based on qualitive analysis. There is not physical or experimental evidence to prove the reasonability of such scoring. For example, the scoring of H/D-Ratio has values ranged from 0.02 to 0.89 and these values are multiplied by 100 to get a 0–100-point scale linearly. In that case, it means that a H/D-Ratio of 0.4 has twice the score (i.e., risk) to a value of 0.2. However, such assumption lacks evidence. No numerical simulation or geological evidence are presented to support the scoring method. This problem also occurs in other four factors.

**Reply: We appreciate this comment and now improve the reasonability and objectiveness of our scoring (F) analysis. We also better discuss the limitations of the approach. As the reviewer correctly points out in the next comment, our score should be seen as an estimation for hierarchy rather than strict empirical criteria.**

**We clarify this by firstly pointing to the purpose of our approach (creating a hierarchy) by adding the following text to the introduction. We make these changes:**

**Page 3, line 79:** "While we incorporate some elements of review studies, which have been done extensively for the volcanic tsunamis in Southeast Asia (see e.g. Paris et al. 2014; Mutaquin et al. 2019), we expand on this by attempting to place the potential source volcanoes in a hierarchical order and identify the most likely volcanoes to cause tsunamis in the future. For this purpose, we create a comprehensive catalogue of potentially tsunamigenic volcanoes and further use this data to create a point-based hierarchical ranking and identify the most likely candidates for sourcing potentially catastrophic tsunamis in the future."

**We then further improve on the method description and reasonability. We note that our score, while being subjective and in lack of empirical hazard relation, is still reasonable to use for creating a hierarchy. We further point out that our data used to create the score is based on very objective criteria that can be measured and quantified. We add this in:**

**Page 6, line 137:** "While there are numerous factors that can be considered to reflect the tsunami hazard from a volcano, most of them do not have a known empirical relation to the hazard. For example, it is reasonable that a steep volcano close to the sea is more likely to produce a tsunami than a gently sloped one far inland, but exactly how much more likely this makes a tsunami is not known. With our ranking, we can therefore only aim to compare these factors by assuming that certain higher values equal a higher hazard. We consider the following five factors and point systems for the ranking. Each represents a set of data that can be recorded or quantified objectively, which is then assigned a subjective but consistent point scale in order to create a comparable hierarchy:"

**Page 13, line 265:** "...our MCDA is based on arbitrary and thus subjective point scales, assigned to best cover the range of values used to build the ranking score."

**Finally, we also appreciate the reviewers comment specifically on the linearity of our scale. Scales like this are usually linear as these are the simplest scales, but can still be reasonable and meaningful. In order to further improve the manuscript, we now have added a discussion paragraph in the limitations section referencing some relevant previous papers as examples. These are the changes:**

**Page 13, line 267:** "In studies previous ranking volcanic hazards and risks, these could be done by a simple count of "yes" or "no" features adding 1 or 0 points, respectively (Yokoyama et al. 1984), or similar variations (Ewert 2007; 2018), or via the creation of index values and adding them up to create a score (Scandone et al. 2016). Then categories (e.g. high, medium and low hazard) are defined to best cover the range of scores (e.g. Ewert 2018), which is also what we do in our ranking. MCDAs in other fields often have more quantitative scales such as 0-9 points (Fernandez et al. 2010; Rahmati et al. 2015) or 0-100 (Nutt et al. 2010), but the score systems are still assigned arbitrarily. Thus, all these approaches and our ranking presented here use some degree of subjective judgement, as not all factors directly translate into an empirical hazard or risk value. Without this, no meaningful comparison between volcanoes could be made. However, as the rules with which points are given are kept strictly the same for all volcanoes, the comparability of scores is retained, allowing for a meaningful hierarchical order or scores. For our ranking, this means that the hazard score by itself should be seen as a rough hierarchy estimation rather than a strict empirical value as it has little meaning in terms of hard data, such as expected tsunami event frequency, possible wave heights, or impacts on shorelines and population. Similarly, we can thus not adequately assess the risk to shores and population in the traditional sense. Instead, we identify which volcanoes are the most likely to cause a tsunami in the future as these are expected to produce the highest hazard score."

2. Similarly, the weighting (W) of the ranking system is also subjective. I agree that the results of robustness testing are satisfactory. But the testing itself cannot show the importance (or contribution) of each factor for MDA. Therefore, the total weighted score can only be used as a rough estimation rather than a strict criterion. The authors may add a confidence level to each total weighted score.

**Reply: We agree that the weights are subjective. We also agree that the weighted score is not a strict criterion. We thus further clarify and improve the subjectivity issue in the methods section by adding the following on:**

**Page 8, line 228:** "For the factor weights, we have to choose values based on the importance of the factor data. A higher weight of a factor will result in a larger impact of this factor on the final score and thus make it more important. Here too, these choices are largely subjective, but allow reducing the impact or importance of e.g. less reliable factor data and in-turn raise the impact of more reliable factors"

**Page 13, line 267-275:** "For our ranking, this means that the hazard score by itself should be seen as a rough hierarchy estimation rather than a strict empirical value as it has little meaning in terms of hard data, such as expected tsunami event frequency, possible wave heights, or impacts on shorelines and population."

**Furthermore, we now follow the advice of the reviewer, by adding standard deviations to quantify this method and modify our robustness testing approach. We show that we can reliably identify the most likely volcanoes to cause future tsunamis despite the subjectivity of our weights, but can be less certain of the sorting with lower scoring volcanoes. We can also identify which volcanoes are more sensitive to the importance (or contribution) of single factors. We have replaced figure 4 and modified our method description and results accordingly.**

**Page 9, line 245:** "We further tested how robust our ranking is with respect to used factor weights. This is done to confirm that the highest scoring volcanoes still retain their high score even when the weighing is significantly different, which can confirm that these volcanoes really pose the highest tsunami hazard despite possible human error or misjudgement. The test was carried out by changing the five factor weights, increasing one factor to 60% and all others are set to 10%. The procedure was repeated for all five factor weights, so that every single factor was once set as the strongest influence. We also added one instance of all weights being considered equal (i.e., all five factors being weighed at 20%). This then enabled us to calculate both an average score and standard deviations within this variability of weights. The results could then be used to judge whether our ranking can generally identify the highest scoring and most hazardous volcanoes well, despite the subjective weight choices. We could further determine which volcanoes were more sensitive to the influence of single factors, as this would result in higher deviations."

**We also updated the figure 4 caption.**

**Page 19, line 357:** "Figure 4: Robustness test of the factor weights used in the ranking. This was done by calculating an average score and standard deviations from repeat scoring while systematically changing the factor weights. It shows that the volcanoes we classed as high hazard volcanoes are generally well distinguished, with the highest values independently of factor weights. This demonstrates that changing the factor weights may slightly change the order in which the volcanoes are ranked, but our analysis is generally classifying higher hazard volcanoes correctly, confirming the robustness of our ranking. However, for the medium and low hazard volcanoes the ranking is less robust, due to a high number of volcanoes with similar scores, which can significantly change the hierarchical order depending on the chosen factor weights."

[Figure]

3. The MDA of the ranking system is based on a linear combination of five individually weighted factors (Equation 1). However, these factors are not mutually independent. For example, a higher slope angle may result in a higher tsunami activity, and therefore, also increases the score of tsunamigenic history. The scoring and weighting of five factors may overlap, which is not appropriate to be represented by a linear combination.

**Reply: We appreciate this comment and improve the manuscript by clarifying the factor dependency. The individually weighted factors we used are, in fact, largely independent. Taking the example above, we actually discuss this for the 2018 Krakatau event at page 22, line 403. The removed steep slope resulted in a lower slope score for Krakatau after the landslide, but a higher tsunami score due to the additional event, so these are measured separately and independently. The only exceptions are the H/D-ratio and slope, which are actually dependent, but this is not problematic for our ranking. We added a statement highlighting this point in the limitations, but use a different example.**

**Page 14 line 290:** "Conducting a comparative ranking can be more challenging if there are major dependencies between the used factors. As an example for our case, it would be reasonable to assume that recent eruptive activity would more likely cause hydrothermal alteration, thus making the eruptive history and hazardous features factors interdependent. However, in our catalogue, only few volcanoes are recorded to have extensive hydrothermal alteration on their flanks and for many of these, no eruption occurred for decades to centuries (e.g. Manuk, Teon, Serua). Hence, we think that these issues are unlikely to significantly affect our results. The only exception is a direct dependence between the H/D-ratio and the slope angle as it is essentially the same value if the volcano is close to the coast, however the separation does allow for a more distinct look at volcanoes that may be far from the coast, but still have steep slopes on a local level."

4. The heat map (Figure 7) and travel-distance plots (Figure 8) cannot accurately represent the potential volcanic tsunami hazards because they do not incorporate the information of tsunami amplitude. It makes the hazard assessment less powerful. A tsunami with 1 m amplitude has evidently different impact from the one with 0.1 m amplitude. I believe it is a MUST to consider the potential maximum amplitude when analyzing volcanic tsunami hazards.

**Reply: We appreciate this comment and make multiple improvements and clarifications to the text and figures, as outlined in the following. Indeed, figures 7 and 8 do not incorporate information on tsunami amplitudes. This is intentional as a reliable assessment of volcano-generated tsunami wave amplitudes requires knowledge of many of yet unknown source parameters. Specifically, there are multiple potential processes at volcanoes which may generate a tsunami (explosion, flank collapse, PDC etc.). Each of them has a specific set of parameters describing magnitude, direction, etc. and each of them would result in highly different wave amplitudes. Reliable modelling of volcanogenic tsunamis requires thorough collection and evaluation of these specific source parameters, in addition to the advanced numerical techniques beyond classical nonlinear shallow water (NLSW) algorithms, and is usually applied to specific singular (historical) events. Incorporating such modelling for multiple volcanoes at once (in a ranking study like present) would not only be highly demanding, but, without constraining all the principal source parameters, also highly speculative.**

**Instead, we would like to avoid producing highly unconstrained results and pursue a simpler and more robust approach by setting our 'tsunami impact metric' to a length of the coastline potentially affected by tsunamis within given propagation time. Note that these simple tsunami travel time models have the advantage that they are independent from the wave height and the generation mechanism (as long as it is a point source), so we can make meaningful assessments without assuming a yet unknown tsunami source.**

**To improve the manuscript, we firstly address this issue by clarifying the aim of the modelling. We particularly emphasise that predictive models (e.g. Giachetti et al. 2012) require in-depth understanding of specific local factors:**

**page 24 line 451:** "Consequently, predictive studies remain rare (Giachetti et al. 2012; Paris et al. 2019) and are only possible because the specific local circumstances leading to the tsunami are very well understood, which is knowledge that is lacking for most coastal volcanoes. Here, we provide multiple predictive models for the volcanoes we classified as posing a high tsunamigenic hazard. As volcanogenic tsunamis are caused by a large variety of mechanisms (Fig. 6) we contribute to this aspect by providing a simplified and broader view at the travel times of potential future tsunamis that are unspecific to the mechanism of tsunami generation and their magnitude (with the possible exception of meteotsunamis as seen at Hunga Tonga Ha'apai in 2022, which appear to have different wave propagation properties). We mainly account for the potential spatial impact of volcanogenic tsunamis and extend our tsunami hazard evaluation by assessing the total length of a coastline affected within one and two hours of tsunami propagation for the volcanoes categorised as high hazard in our ranking (except Didicas)"

**Secondly, we highlight that the amplitudes and wave heights cannot be considered, but that comes with the advantage of the tsunami source independence.**

**Page 24 line 455:** "This means that we can simulate the travel and arrival times of specific volcanoes independent of how the tsunami was generated (as long as it is a point source), but we also cannot consider specific wave heights or runup as these depend strongly on the specific source mechanism and magnitude of the event and require additional and much more specific modelling data for individual sites."

**Page 26 line 475:** "While our models are limited to the travel time, they can be used to estimate the warning time for shores in case a tsunami occurs at one of the considered volcanoes."

**Thirdly, we agree with the reviewer and recognize the value of models with specific wave heights. While we prefer our simplified broader models, we instead provide an additional paragraph summarising some previous studies specific to single volcanoes and historical events:**

**Page 23 line 444:** "In order to assess the risks and impacts of volcanogenic tsunamis, numerical simulations are commonly used, both for distinct future scenarios and in retrospect for past events. For Southeast Asia, a large number of such studies had been conducted. Most models were done for Anak Krakatau looking specifically at the 2018 flank collapse with some using the known event to calibrate and confirm the quality of current simulation methods (Grilli et al. 2019; Borrero et al. 2020; Mulia et al. 2020; Omira and Ramalho 2020; Paris et al. 2020; Zengafinnen et al. 2020), some using the known tsunami data (e.g. from tide gauges) to identify source parameters (Heidarzadeh et al. 2020; Ren et al. 2020; Grilli et al. 2021) and some testing variations in the source parameters to characterise potential future events (Dogan et al. 2021). In general, the consensus is that a landslide between 0.1 and 0.3 km3 volume that occurred both with a subaerial and a submarine component is mostly consistent with the observed and modelled runup heights at the adjacent shores. Similar models also exist for the 1883 tsunami at Krakatau, with the main purpose being the identification of its generation mechanism (Maeno and Imamura 2011) and how such a tsunami propagates in the far-field (Choi et al. 2003). Predictive studies only considering possible future events are not as abundant, but have been done for Anak Krakatau before the 2018 tsunami (Giachetti et al. 2012; Badriana et al. 2017), with Giachetti et al. (2012) making a remarkably close prediction to the later event. Other volcanoes in Southeast Asia are not as commonly considered. Pranantyo et al. (2021) test the tsunami propagation from Ruang volcano, Indonesia, using and comparing both historical observations and data from the 2018 Anak Krakatau event and reproducing

a 25 m runup in the near-field. In Papua New Guinea numerical tsunami models have almost exclusively been considered for the Ritter Island tsunami in 1888 and the reconstruction of its generation (Ward and Day 2003; Karstens et al. 2020). Similarly, numerical tsunami models in the Philippines are mostly limited to Taal volcano, where models are based both on a past tsunami in 1716 (Pakosung et al. 2020) and a predictive study considering scenarios with different explosion sites and energies (Paris et al. 2019). Considering these works, it is clear that tsunamis sourced by volcanoes can be well explained with numerical models, but the considered volcanoes remain limited to a few select sites and scenarios. These models are also typically restricted to one particular volcano and one specific mechanism of tsunami generation as a retrospectively investigation."

**We also make a brief point that our travel-time models could be supplemented with more specific scenario models in future studies.**

**Page 26 line 489:** "For future hazard and risk assessments, we thus recommend supplementing the knowledge from our TTT-models with specific detailed scenario calculations using established numerical modelling approaches, particularly for those high-hazard volcanoes where no such models exist (e.g. Batu Tara, Iliwerung, Nila)."

**Finally, we combined figures 7 and 8 to avoid confusion regarding our TTT models and the heat map highlighting the likely future focus areas for tsunamigenic volcanoes.**

[Figure]

5. The conclusion of this manuscript is too simple. It is necessary to discuss the limitation of this ranking system.

**Reply: We thank the reviewer for raising this point and thoroughly improve the conclusions of the manuscript and now also mention the limitations of the ranking approach. The new conclusion now reads as follows:**

**Page 31 line 607:** "Based on our MCDA analysis considering 131 volcanoes in SE-Asia we identify 19 that pose a high tsunami hazard and another 48 with moderate tsunami hazard. We find our ranking system to be robust for the higher scoring volcanoes, meaning that we can reliably identify the most likely volcanoes to produce a tsunami in the future. For volcanoes with moderate to low scores the ranking is less robust and more susceptible to subjective judgement. The main limitations remaining are (1) a lack of knowledge how much individual factors contribute to the tsunami hazard of a volcano, instead requiring subjective assumptions, (2) erroneous, incomplete or insufficient data availability for many volcanoes (e.g. bathymetry or historical data), and (3) the multitude of different mechanisms which may cause a volcanic tsunami (i.e. PDCs, landslides, explosions), making a clear scenario assessment challenging.

Our results show that the Indonesian Lesser Sunda Islands and northern Molucca Sea as well as the southern Bismarck Sea in Papua New Guinea are areas with a high number of hazardous volcanoes and may thus be particularly prone to tsunamis sourced by volcanoes. Many of these volcanoes such as Batu Tara, Indonesia, are not commonly considered for this type of hazard. We therefore emphasise the need to reconsider the current state of monitoring and risk assessment in these areas. Since tsunami warning systems are mostly not designed to detect volcanogenic tsunamis, our results highlight the importance of a reassessment of the current network and additional suitable equipment on the ground and through earth observation satellites. Due to the inherently short warning times of these events, we also recommended increased pre-emptive measures on a local level, such as increased public education programs for coastal communities and the marking evacuation routes along populated coasts."

Other minor comments:

Line 71: I agree that "the inherent problem of volcanogenic tsunamis is the lack of warning time and quick response options". However, even if we successfully identified the high-tsunami-risk volcanoes, this problem still exists. Please discuss potential solutions (e.g., radar, bottom pressure gauges) to fix this inherent problem.

**Reply: We agree and add a short paragraph. While we cannot fully solve this problem, our work can help prioritise where best to implement these potential solutions.**

**Page 3 line 77:** "This may then allow for a targeted implementation of disaster mitigation strategies and warning systems at critical sites, e.g. by placing additional tide gauges as proposed for Krakatau (Annunziato et al. 2019) or improved volcano monitoring."

**Page 26 line 489:** "For future hazard and risk assessments, we thus recommend supplementing the knowledge from our TTT-models with specific detailed scenario calculations using established numerical modelling approaches, particularly for those high-hazard volcanoes where no such models exist (e.g. Batu Tara, Iliwerung, Nila). This coupled with the prioritisation of specific volcanoes provided by our ranking can provide a well-founded basis for future disaster mitigation strategies. While the detection of volcanic processes triggering a tsunami will remain challenging to detect due to the multiple possible generation mechanisms, other steps can be done to improve warning times. These include the addition of strategically placed tide gauges as suggested by Annunziato et al. (2019), or improved real-time volcano monitoring through seismometers, radar, cameras or infrasound sensors. Regular use of satellite data (e.g. InSAR) can also help to preemptively identify volcanic unrest or destabilising flanks."

Line 105: Add a figure and use an example to show the process of defining the edifice boundary.

**Reply: We agree, but since the edifice boundary is not critical for our scoring, we do this in the supplement and point to it in the text.**

**Page 5 line 110:** "A full example for the NETVOLC and MORVOLC output and an illustration of the edifice boundary definition is provided in the supplementary material A."

Line 332: Remove the repeated word in "high high-hazard".

**Reply: Much appreciated, we corrected this.**

Line 333: Please explain the reason why there are some volcanoes with high scores but not prominently considered for their tsunamigenic potential.

**Reply: This statement is confusingly phrased, we apologise. What we meant to say is that these volcanoes are not as prominently known to be a major tsunami hazard (e.g. Batu Tara, which is not well studied in this regard). We changed the sentence accordingly.**

**Page 15 line 332:** "However, we also identify high-hazard volcanoes that are not as well known for their tsunamigenic potential, but received similarly high scores."

Figure 5: Please add a subpanel to show the respective distribution between the countries for all considered volcanoes.

**Reply: We agree and reworked the figure as suggested, the new subpanel is a)**

[Figure]

Figure 6: What are the different meanings between dark red and light red (also blue, yellow, green, etc.)? Please specify.

**Reply: The dark and light colour are transparency settings mentioned in the figure caption and symbolise that the source mechanism is suspected but uncertain. We clarify this by reworking the figure to a donut plot that we can label more clearly.**

[Figure]

Section 4.3: This section seems verbose. The authors may present Batu Tara here and move others to supplementary material. Instead, it is better to have more discussions on potential tsunami scenarios of Batu Tara.

**Reply: We agree and moved this section as suggested. We now include Batu Tara in the earlier discussion section, including a new figure. The other volcanoes discussion generally serves as a contextualisation of the various high-hazard volcanoes, which may indeed fit best in a supplementary text.**

**Page 23 line 443:** "A brief feature of individual high-hazard volcanoes can be found in the supplementary material C."

---

## Author Comment (AC4)

The manuscript submitted by Dr. Zorn and the colleagues shows the potential of tsunami hazards with volcanic origins in Southeast Asia (Indonesia, Papua New Guinea, Philippines, India, etc.). The authors focused on various factors of 131 volcanoes, such as topographic features, recent volcanic activity, tsunamigenic history in the past, which are considered closely related to tsunami potential, and used a Multicriteria Decision Analysis (MCDA) for the hazard assessment. Then, they found 19 with particularly high tsunami hazard potential, some of which less known and monitored.

While their assessment could not avoid their subjectivity in their definitions of the weights and the points in MCDA, the presented assessment that widely covers major volcanoes in this region is useful to consider tsunami potentials and for further consideration of volcanic tsunami potential at each volcano. I think this manuscript still has some parts to be improved, as listed below, but I believe that this manuscript has the potential to become suitable for publication from NHESS after major revisions.

**Reply: We thank the reviewer for this assessment.**

[Major comments]

1. The objectivity of each factor used in MCDA

In MCDA, the authors considered different factors (H/D-Ratio, Volcanic activity, Tsunamigenic history, Slope angle, and Hazardous Features [Underwater extent, Morphological features, Vegetation, Hydrothermal alteration, Topography between an edifice and the sea]). I suppose that these factors are different in terms of objectivity and uncertainty; in other words, some are objective, while others contain error or subjectivity. For example, H/D-Ratio, slope angle, volcanic activity (if limited to recent activity), and underwater extent are based on rather reliable data. On the other hand, tsunamigenic history should contain many missing events (as the authors mentioned), morphological features cannot be simply quantitatively related to the hazard assessment, the effects of vegetations on edifice stability would depend on their type, etc... I recommend that the authors first use only "more objective" factors, and then add "less objective factors" (at least, please show results only with "more objective" factors, in the supplementary). It would be very helpful for readers' understanding of what are the main factors determining the potential of volcanic hazards.

**Reply: We agree with this comment and now firstly separate both objectivity and subjectivity more clearly:**

**Page 6 line 138:** "We consider the following five factors and point systems for the ranking. Each represents a set of data that can be recorded or quantified objectively, which is then assigned a subjective but consistent point scale in order to create a comparable hierarchy:"

**Secondly, we also comment on the reliability or uncertainty of the factors the reviewer pointed out:**

**Page 8 line 228:** "Morphometry, here meaning H/D-ratio and slope angle, measure both the feasibility of gravitational mass movements (flank collapses or PDCs) reaching the sea, as well as quantify oversteepening of individual flanks. This data also represents the most reliable quantitative data in our ranking as it can be precisely measured."

**Page 8 line 232:** "In turn, we decided to weigh the Hazardous Features less since these are not quantitatively determined and more prone to human subjectivity and misjudgement. Thus they are less reliable."

**Thirdly, we go on to use these separations as a justification for our (still subjective) choices for the factor weights. We can weigh factors less if we do not trust them and give this factor less importance. This is why e.g. the hazardous features were only weighted at 10%. We add a short explanation:**

**Page 8 line 228:** "For the factor weights, we have to choose values based on the importance of the factor data. A higher weight of a factor will result in a larger impact of this factor on the final score and thus make it more important. Here too, these choices are largely subjective, but allow reducing the impact or importance of e.g. less reliable factor data and in-turn raise the impact of more reliable factors. We decided to favour morphometry and eruptive activity over the other factors."

**Finally, for the separate list the reviewer requests here using only reliable factors, we would refer to our interactive excel sheet which can be freely adjusted for that purpose. If e.g. the hazardous features (as the least reliable factor) should be removed, its weight could simply be set to 0% while increasing the others (see instructions on the sheet).**

**Page 15 line 323:** "A complete, more detailed, and interactive version of this list with individual entries relating to how the points were counted can be found in supplementary material B."

2. Similar factors in MCDA

Factors of morphological features and hydrothermal alternation seem to be related to the factor of volcanic activity. It seems that these related factors increase the scores for volcanoes that recently erupted. Please show how these factors are correlated with each other. If the correlations are large, some of the factors might be removed.

**Reply: We fully agree that further clarifications and an improved explanation is needed. The factors we used are, in fact, largely independent and do not correlate on the scoring. We specifically pick up the reviewer example of hydrothermal alteration and volcanic activity and add a statement highlighting this point in the limitations as follows:**

**Page 14 line 290:** "Conducting a comparative ranking can be more challenging if there are major dependencies between the used factors. As an example for our case, it would be reasonable to assume that recent eruptive activity would more likely cause hydrothermal alteration, thus making the eruptive history and hazardous features factors interdependent. However, in our catalogue, only few volcanoes are recorded to have extensive hydrothermal alteration on their flanks and for many of these, no eruption occurred for decades to centuries (e.g. Manuk, Teon, Serua). Hence, we think that these issues are unlikely to significantly affect our results. The only exception is a direct dependence between the H/D-

ratio and the slope angle as it is essentially the same value if the volcano is close to the coast, however the separation does allow for a more distinct look at volcanoes that may be far from the coast, but still have steep slopes on a local level."

3. Potential spatial impact of volcanogenic tsunamis

The map in Fig. 7 does not add any important information, since the heat map of the volcanic tsunamis' spatial impact shows high density around the high-hazard volcanoes, which is obvious. Also, the assessment of the spatial impacts only based on the tsunami travel times is disappointing. To consider the hazard, tsunami amplitudes on coasts should be taken into account. I understand that it is difficult to assume complex volcanic tsunami sources, the authors are recommended to conduct numerical simulations using linear long-wave models, at least with a simple tsunami source model (for example, a Gaussian-shape uplift on the sea surface).

**Reply: We appreciate this comment and make multiple improvements and clarifications to the text and figures. Indeed, we do not include information on tsunami amplitudes or run-up on coasts. This is intentional as a reliable assessment of volcano-generated tsunami wave amplitudes requires knowledge of many of yet unknown source parameters. Specifically, there are multiple potential processes at volcanoes which may generate a tsunami (explosion, flank collapse, PDC etc.). Each of them has a specific set of parameters describing magnitude, direction, etc. and each of them would result in highly different wave amplitudes. Reliable modelling of volcanogenic tsunamis requires thorough collection and evaluation of these specific source parameters, in addition to the advanced numerical techniques beyond classical nonlinear shallow water (NLSW) algorithms, and is usually applied to specific singular (historical) events. Incorporating such modelling for multiple volcanoes at once (in a ranking study like present) would not only be highly demanding, but, without constraining all the principal source parameters, also highly speculative. This also holds true for simpler Gaussian-shape uplifts as the magnitude of uplift would have to be defined based on speculation. Also, a Gaussian-shape source cannot account for any wave directivity which is typical for flank collapses, which is an issue that would become relevant when specific wave heights are considered.**

**Instead, we would like to avoid producing highly unconstrained results and pursue a more meaningful approach by limiting our models to the spatial tsunami extent in time and the length of the potentially affected coast. Note that these simple tsunami travel time models have the advantage that they are independent from the wave height and the generation mechanism (as long as it is a point source), so we can make meaningful assessments without assuming a yet unknown tsunami source.**

**Firstly, we address this issue by clarifying the aim of the modelling. We particularly emphasise that predictive models (e.g. Giachetti et al. 2012) require in-depth understanding of specific local factors::**

**Page 24 line 451:** "Consequently, predictive studies remain rare (Giachetti et al. 2012; Paris et al. 2019) and are only possible because the specific local circumstances leading to the tsunami are very well understood, which is knowledge that is lacking for most coastal volcanoes. Here, we provide multiple

predictive models for the volcanoes we classified as posing a high tsunamigenic hazard. As volcanogenic tsunamis are caused by a large variety of mechanisms (Fig. 6) we contribute to this aspect by providing a simplified and broader view at the travel times of potential future tsunamis that are unspecific to the mechanism of tsunami generation and their magnitude (with the possible exception of meteotsunamis as seen at Hunga Tonga Ha'apai in 2022, which appear to have different wave propagation properties). We mainly account for the potential spatial impact of volcanogenic tsunamis and extend our tsunami hazard evaluation by assessing the total length of a coastline affected within one and two hours of tsunami propagation for the volcanoes categorised as high hazard in our ranking (except Didicas)"

**Secondly, we highlight that the amplitudes and wave heights cannot be considered, but that comes with the advantage of the tsunami source independence.**

**Page 24 line 455:** "This means that we can simulate the travel and arrival times of specific volcanoes independent of how the tsunami was generated (as long as it is a point source), but we also cannot consider specific wave heights or runup as these depend strongly on the specific source mechanism and magnitude of the event and require additional and much more specific modelling data for individual sites."

**Page 26 line 475:** "While our models are limited to the travel time, they can be used to estimate the warning time for shores in case a tsunami occurs at one of the considered volcanoes."

**Thirdly, we agree with the reviewer and recognize the value of models with specific wave heights. While we prefer our simplified broader models, we instead provide an additional paragraph summarising some previous studies specific to single volcanoes and historical events:**

**Page 23 line 444:** "In order to assess the risks and impacts of volcanogenic tsunamis, numerical simulations are commonly used, both for distinct future scenarios and in retrospect for past events. For Southeast Asia, a large number of such studies had been conducted. Most models were done for Anak Krakatau looking specifically at the 2018 flank collapse with some using the known event to calibrate and confirm the quality of current simulation methods (Grilli et al. 2019; Borrero et al. 2020; Mulia et al. 2020; Omira and Ramalho 2020; Paris et al. 2020; Zengafinnen et al. 2020), some using the known tsunami data (e.g. from tide gauges) to identify source parameters (Heidarzadeh et al. 2020; Ren et al. 2020; Grilli et al. 2021) and some testing variations in the source parameters to characterise potential future events (Dogan et al. 2021). In general, the consensus is that a landslide between 0.1 and 0.3 km3 volume that occurred both with a subaerial and a submarine component is mostly consistent with the observed and modelled runup heights at the adjacent shores. Similar models also exist for the 1883 tsunami at Krakatau, with the main purpose being the identification of its generation mechanism (Maeno and Imamura 2011) and how such a tsunami propagates in the far-field (Choi et al. 2003). Predictive studies only considering possible future events are not as abundant, but have been done for Anak Krakatau before the 2018 tsunami (Giachetti et al. 2012; Badriana et al. 2017), with Giachetti et al. (2012) making a remarkably close prediction to the later event. Other volcanoes in Southeast Asia are not as commonly considered. Pranantyo et al. (2021) test the tsunami propagation from Ruang volcano, Indonesia, using and comparing both historical observations and data from the 2018 Anak Krakatau event and reproducing a 25 m runup in the near-field. In Papua New Guinea numerical tsunami models have almost exclusively been considered for the Ritter Island tsunami in 1888 and the reconstruction of its generation (Ward and Day 2003; Karstens et al. 2020). Similarly, numerical tsunami models in the Philippines are mostly limited to Taal volcano, where models are based both on a past tsunami in 1716 (Pakosung et al. 2020) and a predictive study considering scenarios with different explosion sites and energies (Paris et al. 2019). Considering these works, it is clear that tsunamis sourced by volcanoes can be well explained with numerical models, but the considered volcanoes remain limited to a few select sites and scenarios. These

models are also typically restricted to one particular volcano and one specific mechanism of tsunami generation as a retrospectively investigation."

**We also make a brief point that our travel-time models could be supplemented with more specific scenario models in future studies.**

**Page 26 line 489:** "For future hazard and risk assessments, we thus recommend supplementing the knowledge from our TTT-models with specific detailed scenario calculations using established numerical modelling approaches, particularly for those high-hazard volcanoes where no such models exist (e.g. Batu Tara, Iliwerung, Nila)."

**Lastly, regarding the heat-map in Fig. 7, while it may seem obvious, highlights the most likely areas for tsunamis to occur. We think this is important to keep as many of the hazardous volcanoes in the highlighted areas have received little attention and study, which is what we point to with our figure. Here we improve the figure by combining it with Fig. 8 to create a more condensed version and to avoid confusion with our travel-time modelling**

[Figure]

[Minor comments]

Title: As the authors mentioned, submarine volcanoes are not considered in this study. Hence, it would be better to add such as subaerial volcanoes, volcanoes on land, or equivalent words to the title

**Reply: Agreed, we adjust the title accordingly. In SE Asia, there are only 4 known submarine volcanoes which could not be included here, which we also include in the text.**

**The title now reads:** "Identification and ranking of subaerial volcanic tsunami hazard sources in Southeast Asia"

**Page 14 line 287:** "Here, these are Banua Wuhu, Indonesia, and Hankow Reef, Papua New Guinea as well as two unnamed seamounts."

L35: causing some 26% of all volcano induced fatalities

This part is unclear. Do you mean "26% of all volcanoes causing tsunamis"? or 26 % of all volcanoes in the world (irrespective to tsunami generation)?

**Reply: We agree that this statement is confusingly phrased and clarified. Volcanic eruptions cause fatalities by PDCs, lava flows, lahars, etc. and 26% of all are from tsunamis caused by volcanoes.**

**Page 2 line 35:** "...and 26% of all fatalities recorded at volcanoes since 1800 have been due to such tsunamis (Brown et al. 2017)."

L88: although in some circumstances

In what circumstances do volcanoes inland exceed such a distance? Please mention some examples.

**Reply: A very good suggestion, we added some examples in text.**

**Page 3 line 89:** "Deposits of debris avalanches from sector failures of stratovolcanoes, e.g. the Gotemba deposit from Fuji volcano, Japan, were found as far as 24 km from the summit (Yoshida et al., 2012), or 35 km at Shiveluch, Kamtchatka (Belousov et al. 1999). Pyroclastic flows at Mt St. Helens, USA, also reached more than 25 km from the vent (Kieffer, 1981). However, these distances are exceptional and likely limited to very large volume collapses or highly energetic lateral eruptions."

Figure 2:

Here the authors show only a case of Nila volcano in the high hazard category. At least, please show volcanoes in the other two categories for comparison.

**We agree that it is helpful to have a few more examples and have added further figures as suggested. However, because volcanoes in the categories are defined by our scores and not morphological archetypes it would not really be representative to just have one example from either category. Figure 2 mainly served to illustrate the way we measured and recorded our data and Nila happened to have a wide variety of features and thus made for the best showcase example.**

**Instead, we accommodate this suggestion by providing figures with further volcanoes in the discussion and supplement, although focusing more on the high-hazard ones as we consider these to be more important to showcase. Here we added one for Krakatau and Kadovar as a main figure along with Batu Tara. We further present figures of all other high-hazard volcanoes in the supplement, together with the final section of this manuscript, which we also moved to the supplement.**

**Page 23 line 443:** "A brief feature of individual high-hazard volcanoes can be found in the supplementary material C."

**New Figure:**

[Figure]

Anak Krakatau (pre- and post- 2018), Indonesia

Kadovar (pre- and post-2018), Papua New Guinea

Batu Tara, Indonesia

L606– Conclusions

I recommend that the authors add one or a few sentences stating the limitations of this analysis (containing subjectivity and/or errors more or less).

**Reply: We fully agree and thoroughly improve the conclusions of the manuscript and now also state the limitations of this analysis. The new conclusion now reads as follows:**

**Page 31 line 607:** "Based on our MCDA analysis considering 131 volcanoes in SE-Asia we identify 19 that pose a high tsunami hazard and another 48 with moderate tsunami hazard. We find our ranking system to be robust for the higher scoring volcanoes, meaning that we can reliably identify the most likely volcanoes to produce a tsunami in the future. For volcanoes with moderate to low scores the ranking is less robust and more susceptible to subjective judgement. The main limitations remaining are (1) a lack of knowledge how much individual factors contribute to the tsunami hazard of a volcano, instead requiring subjective assumptions, (2) erroneous, incomplete or insufficient data availability for many volcanoes

(e.g. bathymetry or historical data), and (3) the multitude of different mechanisms which may cause a volcanic tsunami (i.e. PDCs, landslides, explosions), making a clear scenario assessment challenging.

Our results show that the Indonesian Lesser Sunda Islands and northern Molucca Sea as well as the southern Bismarck Sea in Papua New Guinea are areas with a high number of hazardous volcanoes and may thus be particularly prone to tsunamis sourced by volcanoes. Many of these volcanoes such as Batu Tara, Indonesia, are not commonly considered for this type of hazard. We therefore emphasise the need to reconsider the current state of monitoring and risk assessment in these areas. Since tsunami warning systems are mostly not designed to detect volcanogenic tsunamis, our results highlight the importance of a reassessment of the current network and additional suitable equipment on the ground and through earth observation satellites. Due to the inherently short warning times of these events, we also recommended increased pre-emptive measures on a local level, such as increased public education programs for coastal communities and the marking evacuation routes along populated coasts."

---

## Author Response (AR2)

**Reviewer 1:**

I appreciate the authors, Dr. Zorn and the colleagues, for their careful consideration on the comments. I am satisfied with the authors' response on my comments including the objectivity of ranking system, weighting, and linear relationship. The quality of figures was also improved. However, the response on the heat map and travel-distance plots is still not clear. Therefore, the reviewer recommends the paper to be accepted for publication after minor revisions have been made to address the following comments.

**Reply: We thank the reviewer for this positive assessment and further address the heat-maps and travel-distance plots, see below.**

(1) I understand the travel and arrival times of specific volcanoes are independent of the magnitude. The authors are pursuing a simpler and more robust approach. However, as a paper on volcano risk assessment, I think it is necessary to incorporate the information of possibility (i.e., hazard score) in the discussion of travel/arrival times. Not all volcanoes have a similar probability to trigger tsunami propagation.

**Reply: We agree with this comment and add this aspect to the discussion as suggested. However, since we are still limited to relative scale of our possibility information from the hazards score, we can only do this qualitatively here. In-turn we emphasize that more specific possibility information (e.g. expected tsunami frequency of individual volcanoes) could be used for a more quantitative assessment using our travel/arrival time data.**

**Page 31, line 605: "Similarly, it is important to emphasise that the probability of the modelled tsunamis are not equal between the volcanoes, and we cannot present a full risk assessment (including e.g. the tsunami probability within a given time period), since the required data is still too sparse. However, our ranking can prioritise which volcanoes are most likely to produce such an event as demonstrated here, but this type of analyses would greatly benefit from quantitative data on future tsunami probability."**

(2) The additional heat map shows the most likely source locations based on a weighted point density calculation using our hazard score (impact multiplied by the hazards score). Please specify the physical meaning of "impact" and show more details about the calculation.

**Reply: We agree and clarify our calculation further. Here, the term "impact" may an unsuitable and confusing word here since we intended it to mean the point density value,**

not a tsunami or volcano impact on the coast. We corrected this and are more explicit on the heat map-calculation.

Page 27, line 517: "For this, we used the kernel density function of Esri®ArcMapTM (version 10.5.0.6492), which calculates the point-density using the interpolated number of points (the volcano locations) within a specified search radius, here ~280 km. The point number is multiplied by our hazard score value for the respective point volcano, which additionally increases the density value in the area around volcanoes with high scores."

(3) The generation mechanism was regarded as a point source. I am wondering whether the assumption could result in some discrepancies in near-field TTT calculation. Please discuss.
Reply: This is a particularly interesting point and we add a short discussion as suggested. Generally this will depend on the triggering mechanism, so for an explosion a point source will likely be appropriate, but a volcanic earthquake may require more complex models

Page 31, line 609: "An additional issue may be the point-based approach of our models, which can cause erroneous arrival times, particularly in the near-field of the tsunami source, if the source is spatially more complex. For volcanic tsunamis with yet unknown properties, point sources are likely appropriate for most cases as the most common sources explosions, PDCs, landslides, and lava dome collapses (Fig. 5) are typically not larger than a few hundred meters. However, large scale sector failures of volcanoes or certain types of earthquakes can likely displace water over larger areas simultaneously and may require more complex source models to accurately capture the tsunami wave near-field."

**Reviewer 2:**
This manuscript of Dr. Zorn and the colleagues has been extensively revised, and the authors excellently responded my and the other reviewer's comments. Their major changes improved the manuscript largely and made the importance of this paper clearer, as follows.
- The authors newly performed a statistical analysis to avoid, as much as possible, the problem regarding the subjectivity and human error, as well as adding detailed and clear discussion parts.

- The authors clearly explained the reason why the tsunami travel time (TTT) modeling is preferred, compared to specific numerical tsunami simulation with a specific source model. Now I agree that the TTT modeling would be a nice approach, given the difficulties, complexities, and variations in modeling volcanic tsunami sources.

- I found that Supplementary File B is useful to examine how the results change when the weights are set differently.

The volcanic tsunami hazard assessments presented in this manuscript that cover volcanoes in this region will give great contributions to the communities' future needs to examine volcanic tsunami potentials at each volcano. I believe that this manuscript will be suitable for publication from NHESS, after modifying a few very minor points below.

**Reply: We thank the reviewer for the positive feedback**

[Minor comments]

Figure 4:

In my understanding, the color is classified by the finally setting of the factor weights, whereas the vertical axis with a plot and an error bar represents the average score and the standard deviations from repeat scoring. If this is correct, please explicitly explain the color is obtained by the final setting of the factor weights in the caption.

**Reply: This is correct, the colors are based on our final setting, while the graph shows the averages and standard deviations. We now clarify this in the figure caption as suggested.**

**Page 22, line 408: "…However, for the medium and low hazard volcanoes (here classified by colour using our final weights and score, see Table 2) the ranking is less robust, due to a high number of volcanoes with similar scores, which can significantly change the hierarchical order depending on the chosen factor weights."**

Also, what the labels a, b and c indicate? Please add the explanation in the caption, or you may remove the labels. The font of Hazard Score in b has a different panel, which should be modified.

**Reply: The a, b, c labels were added as we had to split the plot in 3 separate panels due to the figure size, but as they are still the same plot we agree and removed them here. The fonts in panel b was also adjusted to match the rest of the figure.**